# STRUCTURAL LANGUAGE MODELS FOR CODE GENERATION

## ABSTRACT

We address the problem of *Any-Code-to-Code Generation* (AnyC2C) – generating code given its surrounding code without any restriction on the vocabulary or structure. The state-of-the-art in this problem is the sequence-to-sequence (seq2seq) approach, which treats code as a sequence and does not leverage any structural information. We introduce a new approach to AnyC2C that leverages the strict syntax of programming languages to model a code snippet as a tree – *structural language modeling* (SLM). SLM estimates the probability of the program's abstract syntax tree (AST) by decomposing it into a product of conditional probabilities over its nodes. We present a neural model that computes these conditional probabilities by considering all AST paths leading to a target node. Unlike previous structural techniques that have severely restricted the kinds of expressions that can be generated in this task , our approach can generate arbitrary expressions in any programming language. Our model significantly outperforms both seq2seq and a variety of existing structured approaches in generating Java and C# code. We make our code, datasets, and models available online.

## 1 INTRODUCTION

Generating source code requires reasoning over an unbounded number of syntactic structures and potential symbols. Previous approaches have avoided this issue by limiting the generation problem: program synthesis approaches (Manna and Waldinger, 1971) are tailored to domain-specific languages (Gulwani, 2011), semantic parsing approaches focus on highly templated datasets (Ling et al., 2016; Shin et al., 2019) or SQL (Yu et al., 2018; Dong and Lapata, 2018), while other recent approaches generate code in general languages like Java and C#, but severely restrict the syntax, vocabulary or nature of the generated expressions (Murali et al., 2017; Brockschmidt et al., 2019a).

We introduce the task of *Any-Code-to-Code Generation* (AnyC2C) – generating source code in a general-purpose programming language without any restriction on its vocabulary or structure. Specifically, we focus on generating code in context: given a program $\mathcal{P}$ and some part of the program $p$, predict $p$ from the rest of the program $\mathcal{P}^- = \mathcal{P} \backslash p$. The only restriction we place is that the target $p$ must have a valid subtree within the program's abstract syntax tree (AST). AnyC2C thus generalizes the restricted expression generation task of Brockschmidt et al. (2019a), where target code contains only primitive types and excludes user-defined functions. Figure 1 shows two such AnyC2C examples.

While a sequence-to-sequence (seq2seq) model with a copy mechanism works better than existing code generation approaches on AnyC2C (see Section 5), it ignores the structural information available from the code's AST. We present a new approach that explicitly leverages the strict syntax of programming languages to model code snippets as trees – *structural language modeling* (SLM). SLM estimates the probability of the program's AST by decomposing it into a product of conditional probabilities over its nodes. We present a neural model that computes these conditional probabilities by considering all AST paths leading to a target node. While prior work uses AST paths to *read* programs (Alon et al., 2019a), we *generate* code by producing the target AST node-by-node.

We evaluate SLM on Java AnyC2C benchmarks, where our model achieves a new state-of-the-art exact-match accuracy of 18.04% (previous SOTA: 16.93%). SLMs also outperform existing models on the restricted expression generation task of Brockschmidt et al. (2019a) in C# by a wide margin, 37.61% compared to 26.42%. Our ablation study reveals the importance of using AST paths for

```
public static Path[] stat2Paths(
    FileStatus[] stats) {
  if (stats == null)
    return null;
  Path[] ret = new Path[stats.length];
  for (int i = 0; i < stats.length; ++i) {
    ret[i] = stats[i].getPath() ;
  }
  return ret;
}
```

```
int TrailingSpaces(this StringBuilder builder) {
  var bound = builder.Length - 1;
  if (builder.Length == 0) return 0;
  if (builder[bound] != ' ') return 0;
  var c = 0;
  for (var i = bound; i <= bound; i--) {
    if (i < 0) break;
    if ( builder[i] != ' ') break;
    c++;
  }
  return c;
}
```

Figure 1: AnyC2C examples from the Java (left) and C# (right) test sets. The highlighted expression in each example is the target $p$, which we wish to generate from the rest of the snippet.

both reading and generating code. Finally, we discuss the theoretical advantages of SLMs, and show how they generalize many previous structural approaches for code generation.

## 2 CODE GENERATION AS STRUCTURAL LANGUAGE MODELING

We model the task of Any-Code-to-Code Generation (AnyC2C) by computing the probability of a program $Pr(\mathcal{P})$, similar to how a language model computes the probability of a natural language sentence. While language models typically assume a *sequence* as their input, our input is an abstract syntax *tree* $\mathcal{A}_\mathcal{P}$. We thus introduce a *structural* language modeling approach (SLM) for code generation.

We first show a chain-rule decomposition of the tree's probability $Pr(\mathcal{A}_\mathcal{P})$ into a product of conditional *node* probabilities, and then describe our path-based model for computing the individual conditional probabilities. We explain how to construct a tree from local node predictions, and finally discuss how our approach differs from previous work on production-based tree generation.

**Representing Code as a Tree** A program $\mathcal{P}$ is a sequence of tokens that is unambiguously equivalent to an abstract syntax tree $\mathcal{A}_\mathcal{P}$, where each node represents an element in the language (e.g. conditions, loops, variable declarations) from a set $\mathcal{T}$. The AST's leaves (terminals) have an additional user-defined value $v \in \mathcal{V}$. Nonterminal nodes can have a varying number of children nodes.

**Decomposing the Probability of a Tree** Given a tree $\mathcal{A}_\mathcal{P}$, we first traverse the tree, depth-first,[1] to induce an ordering over its nodes $a_0, \ldots, a_{|\mathcal{A}_\mathcal{P}|} \in \mathcal{A}_\mathcal{P}$. We can now decompose the probability of a tree $Pr(\mathcal{A}_\mathcal{P})$ using the chain rule, akin to the standard approach in language modeling:

$$Pr(\mathcal{A}_\mathcal{P}) = \prod_t Pr(a_t | a_{<t}) \tag{1}$$

where $a_{<t}$ are all the nodes that were traversed before $a_t$.

In AnyC2C, part of the tree ($\mathcal{A}_{\mathcal{P}^-}$) is already observed. Therefore, we order the nodes of $\mathcal{A}_{\mathcal{P}^-}$ before the nodes of the target $p$, and compute only the conditional probabilities over the nodes in $p$, essentially conditioning on the observed tree $\mathcal{A}_{\mathcal{P}^-}$.

**Representing Partial Trees via Paths** How can we represent the partial tree composed of $a_{<t}$ when computing $Pr(a_t | a_{<t})$? In regular language modeling, the structure is linear, and $a_{<t}$ is a sequence. One way to represent a partial tree is to linearize it according to the traversal order (Xiao et al., 2016); however, this could create artificially long distances between the current node $a_t$ and ancestor nodes (e.g., the root $a_0$). Another option is to use only the path from the root node to $a_t$ (Rabinovich et al., 2017), but this ignores a lot of contextual information (e.g., sibling nodes).

We follow Alon et al. (2018) and use *the set of paths* from every leaf to the current node to expand, as well as the path $\mathcal{R}_t$ originating from the root. We denote the (candidate) node at time $t$ as $a_t$, its (given) parent, which is the currently expanded node, by $\pi(a_t)$, and the set of all paths as $\mathcal{S}_t$:

$$\mathcal{S}_t = \{\ell \rightsquigarrow \pi(a_t) \,|\, \ell \in \text{leaves}(a_{<t})\} \cup \{a_0 \rightsquigarrow \pi(a_t)\} \tag{2}$$

---

[1]Depth-first ordering is common practice in tree generation (Maddison and Tarlow, 2014; Raychev et al., 2016; Brockschmidt et al., 2019a), but our framework also allows for other orderings, in theory.

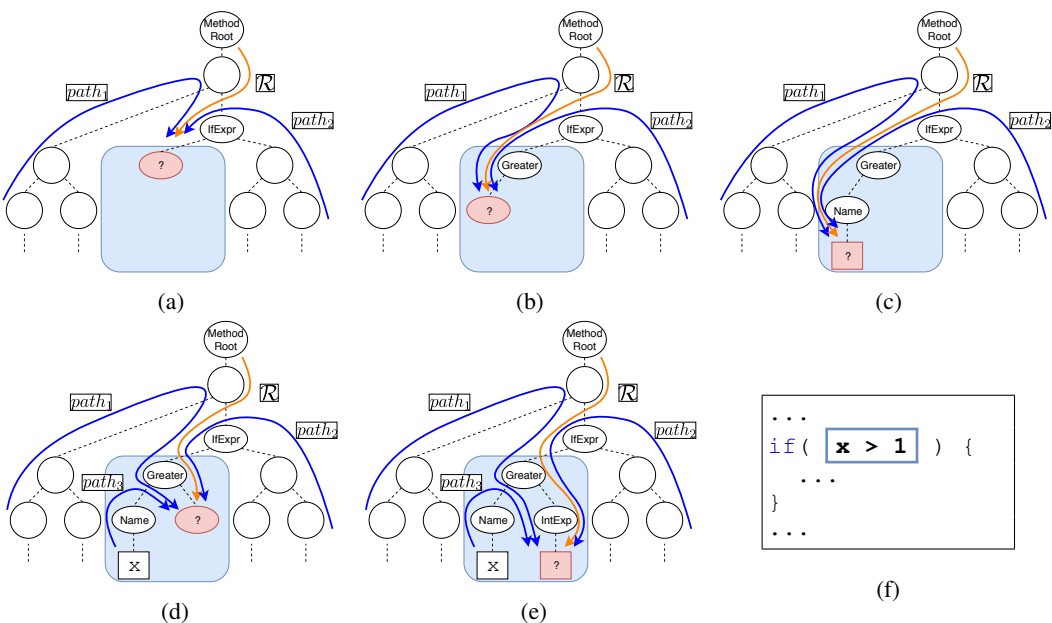

(a)  (b)  (c)

(d)  (e)  (f)

Figure 2: The expression $\boxed{\texttt{x > 1}}$ is generated given its surrounding code context. At each step, the model generates the next node (denoted by a question mark: ⑦) of $path_1$, $path_2$ and $path_3$ using the root path $\mathcal{R}$. Dashed lines denote AST parent-child relations; solid lines denote AST paths.

where $\ell \rightsquigarrow \pi(a_t)$ is the (only) path in the tree between a leaf $\ell$ and the current node to expand $\pi(a_t)$, and $\mathcal{R}_t = a_0 \rightsquigarrow \pi(a_t)$ is the path from the root of the program to $\pi(a_t)$, which represents the current, relative position of $\pi(a_t)$ in the program (marked as $\mathcal{R}$ in Figure 2). Whereas prior work used *whole* paths (between two leaf nodes) to encode an AST (Alon et al., 2019b;a), our model observes *partial* paths (between a leaf and an intermediate node) and learns to extend them.

Figure 2 illustrates the traversal order of a subtree that represents the expression $\texttt{x > 1}$, as well as some of the paths used to compute the probability at each step. At each step, the probability of the next node is computed given the paths $\mathcal{S}_t$ from the root and every given leaf up to the current node to expand. Figure 2(d) shows how after the terminal node $\texttt{Name}$ and its value $\texttt{x}$ are given, $path_3$ originating from this leaf is also used to compute the probability of the next nodes.

Our path-based approach generalizes previous approaches, such as parent feeding (Yin and Neubig, 2017), previous action encoding (Yin and Neubig, 2017), context nodes (Bielik et al., 2016), and some of the graph-edges of Brockschmidt et al. (2019a). See Section 8 for further discussion.

**Generating Trees** In sequence generation, the length of the generated sequence is controlled by generating a single $\texttt{EOS}$ token to stop. When generating trees, we require a more sophisticated mechanism to control arity and depth. We augment $\mathcal{A}_\mathcal{P}$ in two ways to allow node-by-node generation.

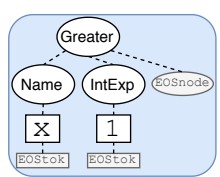

First, we add a special $\texttt{EOS}_{node}$ node to every nonterminal to control for *arity*. Generating this node indicates that the parent node has no more children. Second, we decompose each terminal node $n_v$ into a sequence of terminal nodes $T_v$ by splitting up the node's value $v$ into *subtokens* based on camel notation (Allamanis et al., 2015). For example, if $v = \texttt{toLowerCase}$, then $T_v = \texttt{to} \rightarrow \texttt{lower} \rightarrow \texttt{case} \rightarrow \texttt{EOS}_{tok}$. We end each subtoken sequence with a special $\texttt{EOS}_{tok}$ node to control for *depth* during generation. Figure 3 shows an example of both $\texttt{EOS}_{node}$ and $\texttt{EOS}_{tok}$ in action.

Figure 3: Augmenting the AST with $\texttt{EOS}_{node}$ and $\texttt{EOS}_{tok}$ nodes

**Node Trees vs. Production Trees** While we predict a single *node* at each step, previous work (Iyer et al., 2018; Brockschmidt et al., 2019a) predicts a grammar production rule. This more direct grammatical representation decomposes the code in a way that often forces the model to predict with partial information. For instance, consider the expression $\texttt{str.Substring(3)}$. The

model of Brockschmidt et al. (2019a) would first predict the rule **Expr→Expr.Substring(Expr)**, and only then expand **Expr→str** and **Expr→3**; i.e., the model needs to predict the method name (Substring) *before* the invoking object (str). Further, the Substring method can get either one *or* two arguments, forcing the model to choose whether to use the one- or two-argument production rule in advance. Node generation, however, allows us to predict the presence of a function call and only then to predict its object, method name, and arguments, rather than predicting these a priori.

We note that there exist other approaches that generate an arbitrary number of child nodes with production rule-based models. For example, Rabinovich et al. (2017) used a "horizontal LSTM" to decide whether or not to generate another child; and Yin and Neubig (2018) presented a transition system with a "Reduce" action.

## 3 MODEL ARCHITECTURE

In the previous section, we described how we can generate code given the probabilities $Pr\left(a_t|a_{<t}\right)$, where $a_{<t}$ is represented by the set of partial AST paths $\mathcal{S}_t$. Here, we present a neural model that estimates $Pr\left(a_t|\mathcal{S}_t\right)$. We first apply an LSTM-based path encoder to represent each path in $\mathcal{S}_t$ as a vector (Section 3.1). We then contextualize and aggregate the entire set into a single vector (Section 3.2). Finally, we predict the target node $a_t$ by combining a limited output vocabulary with a syntactic copy mechanism (Section 3.3).

### 3.1 ENCODING AST PATHS

Given a partial AST path (node sequence $n_1, \ldots, n_k$), our goal is to create a vector representation.

We first represent each node $n_i$ using embeddings. A subtoken node is represented by the index of its subtoken $w$ in the embedding matrix $E^{\text{subtoken}}$; AST nodes are represented as a pair $n_i = (\tau, \kappa)$ where $\tau$ is the node type, e.g. IfStatement, and $\kappa$ is the node index among its sibling nodes. We represent node types using a learned embedding matrix $E^{\text{type}}$ and the child indices using a learned matrix $E^{\text{index}}$. The node's vector representation is the concatenation of the type and index vectors.

$$e\left(n_i\right) = \begin{cases} E_w^{\text{subtoken}} & n_i \text{ is the subtoken } w \\ \left[E_\tau^{\text{type}}; E_\kappa^{\text{index}}\right] & n_i \text{ is the AST node } (\tau, \kappa) \end{cases} \tag{3}$$

We encode the entire path using a uni-directional LSTM stack, and take the final states:[2]

$$\widetilde{\overrightarrow{f}}\left(n_1, \ldots, n_k\right) = \text{LSTM}\left(e\left(n_1\right), \ldots, e\left(n_k\right)\right)$$

Given a set of partial paths $\mathcal{S}$ (omitting the iterator $t$ for simplicity), we denote their encodings as $H = \{\widetilde{\overrightarrow{f}}\left(n_1, \ldots, n_k\right) \mid (n_1, \ldots, n_k) \in \mathcal{S}\}$.

**Efficient Computation** When processing an entire tree, there are large overlaps between paths from different time steps. In particular, paths that originate from the same leaf share the same prefix. We therefore apply the LSTM on the prefix *once*, and cache the state across suffixes, speeding up both training and inference significantly. An example is shown in Figure 6 in the Appendix.

### 3.2 AGGREGATING MULTIPLE PATHS

Given the set of paths $\mathcal{S}$ leading up to the target node's parent $\pi(a)$, our goal is to represent $\mathcal{S}$ as a vector, in the context of predicting $a$. To do so, we introduce the aggregation function $g\left(H, r, i\right)$. As its input, $g$ takes the set of encoded paths $H$, the encoded root path $r$, and the child index $i$ of the currently predicted child node $a$ relatively to its parent.

We first contextualize the path encodings $H$ using a transformer encoder (Vaswani et al., 2017).[3] In parallel, we apply a non-linear transformation to the encoding of the root path $r = \widetilde{\overrightarrow{f}}\left(\mathcal{R}\right)$, in order to inform it that we wish to predict the $i$-th child of $\pi(a_t)$:

$$Z = \text{Transformer}\left(H\right) \qquad\qquad \widetilde{r} = W_a \cdot \text{ReLU}\left(C_i \cdot r\right) \tag{4}$$

---

[2]Replacing the LSTMs with transformers yielded similar results in preliminary experiments.

[3]Since $H$ is a set, we do not use positional embeddings.

In this formulation, the parameter matrix $C_i$ is used when the child index is $i$, while the parameter matrix $W_a$ is used for every instance.

We then compute attention over the set of contextualized path encodings $Z$ with the index-informed root-path encoding $\widetilde{r}$ as the query, pass the weighted average $\widetilde{z}$ and the root-path encoding $\widetilde{r}$ through another fully-connected layer and denote the resulting vector representation as $\widetilde{h}$:

$$\boldsymbol{\alpha} = \mathrm{softmax}\left(Z \cdot \widetilde{r}\right) \qquad \widetilde{z} = \sum_j \alpha_j \cdot Z_j \qquad \widetilde{h} = g\left(H, r, i\right) = \mathrm{ReLU}\left(W_g\left[\widetilde{z}; \widetilde{r}\right]\right) \quad (5)$$

### 3.3 PREDICTING WITH A SYNTACTIC COPY MECHANISM

We can now predict $a$ from the representation $\widetilde{h}$. If the target node's parent $\pi(a)$ is a nonterminal AST node, then $a$ must be an AST node; otherwise, $a$ is a subtoken.

**Predicting AST Nodes** We predict $a$ using a softmax over the node type embeddings $E^{\mathrm{type}}$:

$$Pr\left(a|\mathcal{S}\right) = \mathrm{softmax}\left(E^{\mathrm{type}} \cdot \widetilde{h}\right) \qquad (\pi(a) \text{ is a nonterminal}) \qquad (6)$$

**Predicting Subtokens** Programs repeatedly refer to previously declared symbols, resulting in highly repetitive usage of identifiers. We therefore add a copy mechanism (Gu et al., 2016) to allow our model to predict either entire tokens or subtokens that already exist in the context. As we show in Section 6, copying greatly improves our model's performance. For brevity, we describe how entire tokens are copied, and elaborate on the copy of subtokens in Appendix C. We score each leaf $\ell$ using a bilinear function ($W_c$) between its path's encoding $H_\ell$ and $\widetilde{h}$. At the same time, we score the token $w$, which is the token associated with $\ell$, from a limited vocabulary using the inner product between its representation in the subtoken embedding matrix $E^{\mathrm{subtoken}}$ and $\widetilde{h}$.

$$s_{\mathrm{copy}}\left(\ell\right) = H_\ell \cdot W_c \cdot \widetilde{h} \qquad\qquad s_{\mathrm{gen}}\left(w\right) = E_w^{\mathrm{subtoken}} \cdot \widetilde{h} \qquad (7)$$

The scores $s_{\mathrm{copy}}$ and $s_{\mathrm{gen}}$ are then summed over different occurrences that correspond to the same symbol, and subsequently normalized via softmax. A key difference from previous work (Gu et al., 2016; Yin and Neubig, 2017) is that our copy mechanism uses the *syntactic* relation between the source and the target (AST path), rather than their sequential relation. Yin et al. (2019) proposed a graph-based copying mechanism that is capable of copying both tokens and subtrees from the context.

## 4 EXPERIMENTAL SETUP

### 4.1 BENCHMARKS

**Any-Code Generation (AnyC2C): Java** We take the Java-small dataset of Alon et al. (2019a), which contains 11 GitHub projects, broken down to a single method per example, and split to train/dev/test by project to reduce code overlap. This dataset was found to contain the least code duplication by Allamanis (2018b). We create AnyC2C examples by selecting every expression larger than a single AST node as the target, using the remainder of the method as the context. We remove methods that contain the word "test" in their body or file name, and remove methods longer than 20 lines to avoid auto-generated code. To make the task even harder, we remove examples where the target subtree appear as-is in the context. This dataset contains 1.3M/10k/20k train/dev/test examples. The average number of targets for our model is 10.8; for the seq2seq baselines the average is 7.8 targets; if we modeled our targets using production rules, the average would have been 7.9.

**Restrict Code Generation (RestrictC2C): C#** Since the restricted expression generation (RestrictC2C) dataset of Brockschmidt et al. (2019a) is not publicly available, we consulted with Brockschmidt et al. directly and use the dataset of Allamanis et al. (2018a). This dataset contains 30 GitHub projects broken down to one method per example, and use the "unseen projects test" split. To create RestrictC2C examples, we use the code of Brockschmidt et al. (2019a) which filters out examples where the targets contain non-primitive types or user-defined functions. We extract the exact same types of limited expressions. This dataset contains 16k/8k/3k train/dev/test examples.

Detailed statistics of all datasets are provided in Appendix A.

**Metrics** We follow Brockschmidt et al. (2019a) and report exact match accuracy at 1 and 5. We also introduce a new *tree@k* metric, which counts a prediction as correct if the entire tree structure, ignoring leaf values, is identical to the tree of the ground truth. For example, the expressions `x > 1` and `y > 2` would *not* count as identical in exact match, but *would* count as "tree-match identical" because both express that an identifier is greater than an integer (`NAME > INT`). *tree@k* is interesting because it allows us to tease apart the model's syntactic errors from incorrect subtoken predictions.

## 4.2 BASELINES

We compare our model to a variety of original implementations and adaptations of existing models. We put significant effort to perform a fair comparison, including adding a copy mechanism to the NMT baselines and *sub*tokenization as in our model. We adapt strong baselines from the literature to our task, even if they were designed to different tasks such as NL→code and code→NL. We re-train all the following baselines on the same datasets as our model.

**Neural Machine Translation** We use standard autoregressive sequence-to-sequence NMT baselines, in which we subtokenize the given code snippet, replace the target in the source with a special `PRED` symbol, and train the network to predict the target as a sequence of subtokens. *Transformer-base+copy* (Vaswani et al., 2017) uses the implementation of OpenNMT (Klein et al., 2017) with a copy mechanism (Gu et al., 2016). *Transformer-small+copy* uses $d_{\text{model}} = 256$, $d_{\text{ff}} = 1024$, and 4 self attention heads per layer. *BiLSTM→LSTM+copy* is a $d = 512$ 2-layer bidirectional LSTM encoder-decoder with attention (Luong et al., 2015). *seq2tree+copy* follows Aharoni and Goldberg (2017) and learns to generate the linearized, subtokenized target AST, with the same architecture as *BiLSTM→LSTM+copy*.

**Java-specific Baselines** We used the original implementation of Iyer et al. (2018), and also their *seq2prod* baseline which is a re-implementation of Yin and Neubig (2017); these are designed for NL→code tasks, in which we feed the (sub)tokenized code context as the NL input. The model of Iyer et al. (2018) is designed to get additional input of the available variables *and their types*, for which we do not feed types. While in theory these models could also be applied to other languages, their implementation only supports Java.

**C#-specific Baselines** We compare our model to $GNN{\rightarrow}\mathcal{NAG}$ using the original implementation of Brockschmidt et al. (2019a) which contains additional improvements using ideas from Cvitkovic et al. (2019). Bielik et al. (2016) kindly trained and tested their non-neural PHOG model on our C# dataset. We note that PHOG does not have an explicit copy mechanism, and considers only context to the left of the target code, while we consider also context to the right. Extending PHOG to use copying and considering more context could potentially improve its results.

In both Java and C#, we compare to *code2seq* (Alon et al., 2019a), which is a strong code→NL model and train it to generate the target code as a *sequence* of subtokens.

## 4.3 IMPLEMENTATION AND HYPERPARAMETER SETTINGS

**Architecture** We use embeddings of size 512, 2 layers of LSTMs with 256 units, and 4 transformer layers with 8 attention heads. We kept a subtoken vocabulary of size 1,000 to encourage the model to learn to copy; larger vocabularies did not show an improvement. These resulted in a very lightweight model of only 15M trainable parameters, which is close to *Transformer-small* (11.8M parameters). In comparison, the *Transformer-base* model had more than 45M trainable parameters.

**Training** We train the model end-to-end using the cross entropy objective and the Adam optimizer (Kingma and Ba, 2014), an initial learning rate of $10^{-4}$ decayed by a factor of $0.95$ every $20k$ steps. We bucket examples based on the number of predictions in the target subtree (nodes + subtokens + `EOS` symbols), and vary the batch size such that each batch contains about $512$ targets. We train the model to prefer copying entire tokens rather than copying subtokens, if possible. We apply dropout of $0.25$ in the Transformer layers, and a recurrent dropout of $0.5$ in the LSTMs.

**Inference** We perform beam search with width of 5, and optimize for accuracy@1.

| Model | acc@1 | acc@5 | tree@1 | tree@5 |
|---|---|---|---|---|
| code2seq (Alon et al., 2019a) | 10.68 | 15.56 | 30.46 | 43.94 |
| Iyer et al. (2018) | 5.94 | 9.19 | 25.54 | 36.75 |
| seq2prod (Yin and Neubig, 2017) | 8.05 | 11.82 | 30.77 | 41.73 |
| Transformer-small (Vaswani et al., 2017)+copy | 14.23 | 21.35 | 31.83 | 47.40 |
| Transformer-base (Vaswani et al., 2017)+copy | 16.65 | 24.05 | 34.68 | 50.52 |
| BiLSTM→LSTM (Luong et al., 2015)+copy | 16.93 | 23.17 | 34.29 | 49.72 |
| seq2tree (Aharoni and Goldberg, 2017)+copy | 16.81 | 23.04 | 38.14 | 52.36 |
| **SLM (this work)** | **18.04** | **24.83** | **39.10** | **55.32** |

Table 1: Results on Any-Code-to-Code Generation (AnyC2C) in Java.

| Model | acc@1 | acc@5 | tree@1 | tree@5 |
|---|---|---|---|---|
| $GNN{\rightarrow}\mathcal{NAG}$ | 15.19 | 27.05 | 26.48 | 40.09 |
| code2seq | 6.20 | 10.05 | 21.97 | 30.89 |
| seq2seq+copy | 26.42 | 37.94 | 34.10 | 49.23 |
| seq2tree+copy | 22.29 | 35.86 | 31.85 | 48.53 |
| PHOG | 7.40 | 12.00 | – | – |
| **SLM (this work)** | **37.61** | **45.51** | **51.10** | **59.82** |

| Ablation | acc@1 | acc@5 |
|---|---|---|
| Paths→Seq | 12.95 | 18.52 |
| Seq→Path | 12.12 | 17.12 |
| Paths→Paths | 17.63 | 24.62 |
| No Root Att | 14.43 | 18.48 |
| No Copy | 10.72 | 15.70 |
| SLM (original model) | 18.04 | 24.83 |

Table 2: Results on RestrictC2C in C#.    Table 3: Ablations on AnyC2C in Java.

## 5 Results

**Any-Code Generation: Java** Table 1 shows that our SLM achieves over $1.1\%$ and $0.78\%$ better *acc@1* and *acc@5* (respectively) over the two strongest baselines. The improvement over *Transformer-small*, which is closer to our model in the number of parameters, is even higher: over 3.8% and 3.4% in *acc@1* and *acc@5*.

In general, the NMT baselines performed better than code-specific baselines. We hypothesize that the reason is that the NMT baselines are more generic, while the code-specific baselines are designed for different tasks: *seq2prod* is designed for tasks which involve generating code *given natural language input*; Iyer et al. (2018) additionally expects all member methods, variables, and their types as input; *code2seq* is designed to generate sequences rather than code, and does not have a copy mechanism. An approximation of *code2seq* with a copy mechanism is presented in Section 6.

Interestingly, the syntactically-informed *seq2tree* baseline achieved the highest *tree@k* among the baselines, while our model achieved higher *acc@k* and *tree@k*. This shows that leveraging the syntax can be beneficial in NMT baselines as well.

**Restricted Code Generation (RestrictC2C): C#** Table 2 shows the results for the RestrictC2C task in C#, where *seq2seq+copy* is the *BiLSTM→LSTM+copy* model which performed the best among the Java baselines. We first observe that the *seq2seq+copy* and the *seq2tree+copy* baselines outperform the $GNN{\rightarrow}\mathcal{NAG}$ of Brockschmidt et al. (2019a), who introduced this task. Although Brockschmidt et al. (2019a) did compare to a seq2seq baseline, their $GNN{\rightarrow}\mathcal{NAG}$ model could copy symbols from the context, but their baseline did not. To conduct a fair comparison with our SLM model, we equip the seq2seq and seq2tree baselines with a copy mechanism. Even though the *seq2seq+copy* and the *seq2tree+copy* baselines perform substantially better than the state of the art in this setting, our SLM model is able to go beyond, achieving significant gains over all models.

Examples for predictions made by our model and baselines can be found in Appendices D and E.

## 6 Ablation Study

To understand the importance of the various components and design decisions in our model, we conducted an extensive ablation study on the AnyC2C task in Java.

```
private void handleTaskFinishedEvent(TaskFinishedEvent event) {
  TaskInfo taskInfo = info.tasksMap.get( event.getTaskId() );
  taskInfo.counters = event.getCounters();
  taskInfo.finishTime = event.getFinishTime();
  taskInfo.status = TaskStatus.State.SUCCEEDED.toString();
  taskInfo.successfulAttemptId = event.getSuccessfulTaskAttemptId();
}
```

| | | | |
|---|---|---|---|
| True ref: | `event.getTaskId()` | | |
| | `event.getTaskName()` | (8.8%) | (tree-match) |
| SLM top-5 candidates: | `event.getId()` | (8.2%) | (tree-match) |
| | `event.getTask()` | (3.4%) | (tree-match) |
| | `event.getName()` | (3.3%) | (tree-match) |
| | **`event.getTaskId()`** | (3.3%) | (exact match) |

Figure 4: A Java AnyC2C example from our test set along with the predictions of our model. The predictions of the baselines are shown in Figure 8 in Appendix D.

***Paths→Seq*** follows *code2seq* (Alon et al., 2019a) and separates the model to an encoder and a decoder, where the decoder generates the target subtree as a sequence of subtokens. The main difference from *code2seq* is that *Paths→Seq* includes a copy mechanism, as in our SLM model.

***Seq→Path*** follows Rabinovich et al. (2017) and separates the model to an encoder and a decoder (including a copy mechanism), where the encoder encodes the context as a sequence of subtokens using a BiLSTM, and the decoder uses only the root path and the index of the generated child.

***Paths→Paths*** uses separate encoder and decoder which both are AST-path based. These encoder and decoder have different parameters, unlike our SLM model which models the context and the prediction using the same components.

***No Root Attention*** uses max pooling instead of attention in aggregating multiple paths (see Section 3.2). The index-informed path from the root to the target's parent ($\mathcal{R}$ in Figure 2) is concatenated with the result, instead of being used as attention query.

***No Copy*** replaces copy mechanism with a much larger vocabulary (25k subtokens instead of 1k).

Table 3 shows the results of these alternatives. The significantly lower results of *Paths→Seq* and *Seq→Path* show the great benefit of using a unified *structural language model*, instead of separate encoder and decoder components. While this separation between encoders and decoders might be necessary in semantic parsing (Rabinovich et al., 2017; Dong and Lapata, 2018), NL→code (Yin and Neubig, 2017) and code→NL (Alon et al., 2019a; Fernandes et al., 2019) tasks because of the different modalities of the input and the output, this separation may hurt performance when the output is essentially a missing part of the input's AST. As expected, *Paths→Seq* performs better than *code2seq* (Table 1), as it includes a copy mechanism and *code2seq* does not.

As SLM performs better than *Paths→Paths*, this ablation shows the importance of joint modeling of the context and the target subtree by parameter tying. Each of *Paths→Paths* and the seq2seq baselines (Table 1) performs better than *Paths→Seq* and *Seq→Path*; this shows the importance of *using the same type of encoder and decoder* for the AnyC2C task, rather than combining "an optimal encoder" with "an optimal decoder". *Paths→Paths* performs better than the seq2seq baselines (Table 1), showing the advantage of using paths over textual sequences, even without parameter tying.

*No Root Attention* degrades *acc@1* and *acc@5* by 3.6% to 6.3%. This shows that dynamically attending to the context paths given the current root path is crucial, even though the root path is necessarily included as a sub-path of other paths in the set $\mathcal{S}_t$ which go through self-attention.

*Not using a copying mechanism* results in a degradation of 7.3% to 9.1%. Programs use symbols and identifiers repetitively, thus the ability to copy symbols from the context is crucial for this task. For this reason, we included a copying mechanism in all NMT baselines in Section 4.

```
protected void checkRpcAdminAccess() throws
    IOException, AccessControlException {
  UserGroupInformation ugi = UserGroupInformation.getCurrentUser();
  UserGroupInformation zkfcUgi = UserGroupInformation.getLoginUser();
  if (adminAcl.isUserAllowed(ugi) ||
    ugi.getShortUserName().equals( zkfcUgi.getShortUserName() )) {
      LOG.info("Allowed RPC access from " + ugi
        + " at " + Server.getRemoteAddress());
      return;
    }
  String msg = "Disallowed RPC access from " + ugi
    + " at " + Server.getRemoteAddress()
    + ". Not listed in " + DFSConfigKeys.DFS_ADMIN;
  LOG.warn(msg);
  throw new AccessControlException(msg);
}
```

| True ref: | `zkfcUgi.getShortUserName()` | | |
|---|---|---|---|
| | **`zkfcUgi.getShortUserName()`** | (11.7%) | (exact match) |
| SLM top-5 candidates: | `DFSConfigKeys.DFS` | (4.5%) | |
| | `zkfcUgi.getUserName()` | (2.6%) | (tree-match) |
| | `zkfcUgi.getUser()` | (1.7%) | (tree-match) |
| | `zkfcUgi.getUserId()` | (0.6%) | (tree-match) |

*Entirely copied tokens* are marked in brown; unknown *copied subtokens* are marked in blue; *in-vocabulary* subtokens are marked in black; subtokens that are *both in-vocabulary and copied* from context are marked in purple.

Figure 5: A Java AnyC2C example from our test set along with the predictions of our model. The predictions of the baselines are shown in Figure 7 in Appendix D.

## 7 QUALITATIVE ANALYSIS

### 7.1 CORRECT TREE, INCORRECT IDENTIFIER ASSIGNMENT

As shown in Section 5, there is a gap between *acc@k* and *tree@k* across all models: when ignoring identifier values and considering only the tree structure, the accuracy is significantly higher. Our SLM model performs better than all baselines in *acc@k* (Table 1); further, our model also shows a greater potential for improvement in its *tree@k* results which are much higher than the baselines.

We focus on the studying the cases where the tree structure was predicted correctly, but the model failed to generate the code exactly including names. Figure 4 shows a representative example for this case: the ground truth `event.getTaskId()` was predicted correctly only as the fifth candidate; nevertheless, all top-5 candidates are a "tree-match" since all of them express a method which is invoked on an object without arguments, of the form: `NAME.NAME()`. Generating the correct method name `[get,task,id]` is very difficult in this case, since neither `getTaskId` nor `TaskId` appear in the context and there is no apparent hint for them.

### 7.2 USEFULNESS OF COPY MECHANISM

As shown in Section 6, the ability to copy is crucial for the AnyC2C task, because of the repetitive use of identifiers and symbols in programs. Figure 5 shows a representative example for the necessity of the copy mechanism: generating the ground truth `zkfcUgi.getShortUserName()` is feasible *only* thanks to the copy mechanism, since `zkfc` is obviously an UNK subtoken which was not observed in the training data.

In this case, since both `zkfcUgi` and `getShortUserName` appear in context, both were copied as *entire tokens*, rather than generated using subtokens. This example also shows how the ability to copy *entire tokens* ease the generation process by reducing the number of target symbols (our SLM model is able to copy and combine single subtokens as well).

## 8 RELATED WORK

**Generalizing Previous Approaches** Our approach frames code generation as predicting the next node in all partial AST paths. This simple framing generalizes most previous work, without hand-crafted special edges and actions:

- All models that use information about ancestor nodes only (Rabinovich et al., 2017; Maddison and Tarlow, 2014), as well as the "Parent Feeding" of Yin and Neubig (2017), are generalized by our model, since all paths that go into a node $a_t$ pass through its parent, and the path from the root $\mathcal{R}_t$ (Figure 2) is used as the attention query.
- The "previous action encoding" of Yin and Neubig (2017) is also a special case of our approach, because $\mathcal{S}_t$ contains the paths starting from the *previously expanded* leaves of $\mathcal{A}_p$ into the currently expanded node $\pi(a_t)$, such as $path_3$ in Figure 2(e).
- The "context node" of PHOG (Bielik et al., 2016) is just one of the previously-traversed leaf nodes in $a_{<t}$. Thus, not only that our model conditions on this context node as well, our model also takes into account the *syntactic relation*, i.e., the path, between the context and $\pi(a_t)$. Moreover, while PHOG conditions on a single leaf, SLMs condition on *every* leaf in $a_{<t}$.
- Finally, Brockschmidt et al. (2019a) define special graph edges (e.g., "NextSib" and "Child") to capture relations on the AST. Most of these relations can be expressed as partial AST paths.

**Program Generation** Learning to generate programs is one of the oldest problems in machine learning (Waldinger and Lee, 1969) and has been considered by some as the "holy grail of computer science" (Pnueli and Rosner, 1989; Gulwani et al., 2017). Typically, the task is to generate a program given some form of input or context, such as complete formal specifications (Green, 1981) or input-output examples (Gulwani, 2011; Devlin et al., 2017; Parisotto et al., 2017). While these approaches work well in some cases, they are bounded to DSLs that prevent them from being applied to realistic, general-purpose code. Maddison and Tarlow (2014) and Amodio et al. (2017) generate supposedly general-purpose code in a modern programming language, but do not deal with the challenge of fitting the code to a given context. Murali et al. (2017) generate code conditioned on a set of APIs; they state that their approach is thus intrinsically limited to generate API-heavy programs and is unusable for general, logical programs lacking external calls. Further, their generated programs are in an only "Java-like" language. Yin and Neubig (2017), Iyer et al. (2018) and Rabinovich et al. (2017) generated general-purpose code as well, but for another task of generating code given natural language description. Yin et al. (2019) generated "any code" given an encoded edit that needs to be applied to a given code snippet.

Other work used datasets that are either small (Ling et al., 2016), containing highly aligned examples (Oda et al., 2015; Chen et al., 2018), limited-purpose languages like SQL (Yu et al., 2018), or general-purpose but containing eminently templated programs (Ling et al., 2016). Brockschmidt et al. (2019a) limit their model to generate only expressions of primitive types or arrays of these; use a closed vocabulary; and ignore expressions containing user-defined functions, because function names are hardcoded in their syntax production rules. In this paper, we lift these constraints and allow any, general-purpose, generation of code, of all types and containing any names. Our work is also related to Habash (2004), who used structural n-grams over dependency trees for statistical machine translation (SMT).

## 9 CONCLUSION

We presented a novel approach for generating code given surrounding context: computing the probability of an AST using a structural language model. We show that our approach generalizes most previous work in this area, while reaching state-of-the-art performance on of challenging benchmarks. We are eager to see future work advance SLMs further, and apply them to other real-life coding applications as well as other structured-data domains.

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

Table 4: Statistics of our datasets. When not mentioned otherwise, the statistic was measured on the training set.

|  | Java | C# |
|---|---:|---:|
| #projects - training | 9 | 25 |
| #projects - dev | 1 | 2 |
| #projects - test | 1 | 3 |
| #examples - training | 1,309,842 | 16,295 |
| #examples - dev | 10,000 | 8,183 |
| #examples - test | 20,000 | 3,305 |
| Avg. number of paths | 27.8 | 131.1 |
| Avg. source length - lines | 10.4 | 57.5 |
| Avg. source length - tokens | 77.7 | 264.3 |
| Avg. source length - subtokens | 100.6 | 343.6 |
| Avg. target length - tokens | 5.4 | 3.9 |
| Avg. target length - subtokens | 7.8 | 5.0 |
| Avg. target length - tree nodes | 3.8 | 3.9 |
| Avg. target length - tree targets (including subtokens & EOS) | 10.8 | 10.8 |

## A   DATA STATISTICS

Table 4 shows some statistics of our used datasets. In Java: for the validation set, we randomly sampled $10,000$ examples from the raw dev set; for the test set, we randomly sampled $20,000$ examples from the raw test set.

## B   CODE GENERATION PSEUDOCODE

Algorithm 1 shows the pseudocode for our depth-first, left-to-right, code generation approach. We keep a stack (line 1) which is initialized with an initial node to expand. We loop while the stack is not empty (line 3), and pop (line 4) the next node to expand at each step. If the node to expand is a nonterminal, we predict a child *node* (line 7). If the node is a terminal to a subtoken, we predict a subtoken from a vocabulary or copy (line 7). If the predicted child node, whether AST node or subtoken, can be further expanded (line 10), it is pushed back to the stack (line 12).

## C   COPYING SINGLE SUBTOKENS

In addition to scoring the entire token to be copied, we also score each of the subtokens composing it according to their position. For each position $i$, we add a scoring function $s_{copy_i}$, such that $s_{copy_i}(\ell)$ produces the copying score of the $i$'th subtoken of $\ell$, which we denote as $\ell_i$:

$$s_w = s_{\text{gen}}(w) + \sum_{\text{val}(\ell)=w} s_{\text{copy\_token}}(\ell) + \sum_{i} \sum_{\text{val}(\ell_i)=w} s_{\text{copy}_i}(\ell) \tag{8}$$

$$Pr(a|\mathcal{S}) = \text{softmax}(s) \tag{9}$$

Where $s_{\text{copy\_token}}$ is the scoring function of copying the entire token, described in Section 3.3.

For example, a token of `getX` is scored entirely using $s_{\text{copy\_token}}$; each of its subtokens, `get` and `X`, are scored using $s_{copy_1}$ and $s_{copy_2}$ respectively. That is, the model can either copy the entire token, or copy only some of its subtokens. This ability is especially useful in generating a name like `setX`, where `getX` appears in the context, and `X` is any unknown, user-defined, subtoken; the model learns to generate `set` from the vocabulary, and copy only the subtoken `X`.

## D   JAVA EXAMPLES

Figures 5-12 contain examples from our test set for the AnyC2C task in Java, along with the prediction of our model and some of the baselines. The highlighted expressions are the true references that should be generated. Indentation and line breaks may have been altered for typesetting reasons.

---

**Algorithm 1:** Pseudocode for the code generation algorithm.

---

**Input** : partial AST $\mathcal{A}^-$, initial node to expand $a_0$
**Output:** subtree $\tau$

1   $stack \leftarrow$ emptyStack(); push($stack, a_0$);
2   $\tau \leftarrow$ initTree($a_0$) ;
3   **while** $stack$ is not empty **do**
4     $parent \leftarrow$ pop($stack$);
5     $\mathcal{S} \leftarrow$ encodePaths($\mathcal{A}^-, \tau$, parent);
6     **if** $parent \in nonterminals$ **then**
7       $a \leftarrow$ predictNode($\mathcal{S}, parent$);
8     **else if** $parent \in terminals$ or $parent \in subtokens$ **then**
9       $a \leftarrow$ predictSubtoken($\mathcal{S}, parent$);
10     **if** $a$ is not EOS **then**
11       $\tau \leftarrow$ insertChild($\tau, parent, y$);
12       push($stack, a$);
13   return $\tau$;

---

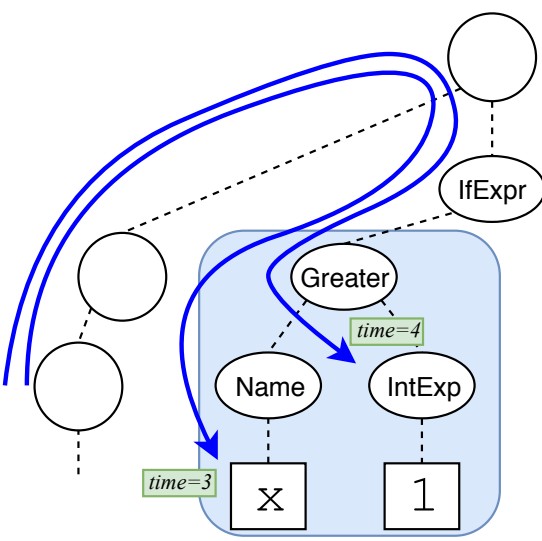

Figure 6: Efficient computation: partial paths for different time steps share the same prefix, allowing a shared computation. In this example, the prefix is the shared path from the leaf (not shown) to Greater, and is much longer than either of the suffixes.

# E   C# EXAMPLES

Figures 13-20 contain examples from our test set for the RestrictC2C task in C# along with the prediction of our model some of the baselines. The highlighted expressions are the true references that should be generated. Indentation and line breaks may have been altered for typesetting reasons.

```
private static String getNameServiceId(
    Configuration conf, String addressKey) {
  String nameserviceId = conf.get(DFS_NAMESERVICE_ID);
  if (nameserviceId != null) {
    return nameserviceId;
  }
  Collection<String> nsIds = getNameServiceIds(conf);
  if (1 == nsIds.size() ) {
    return nsIds.toArray(new String[1])[0];
  }
  String nnId = conf.get(DFS_HA_NAMENODE_ID_KEY);
  return
    getSuffixIDs(conf, addressKey, null, nnId, LOCAL_ADDRESS_MATCHER)[0];
}
```

| Model | Predictions | |
|---|---|---|
| True ref: | `nsIds.size()` | |
| SLM (this work) | **`nsIds.size()`** | (83.7%) |
| | `conf.size()` | (3.0%) |
| | `getSuffixIDs(conf).length` | (2.5%) |
| Transformer-base +copy | `-1` | |
| | `ns.size()` | |
| | `conf.size()` | |
| BiLSTM→LSTM +copy | `-1` | |
| | `Integer.MAX_VALUE` | |
| | `conf.size()` | |
| Seq2tree +copy | `1` | |
| | **`nsIds.size()`** | |
| | `stringPool.blank` | |

```
protected void checkRpcAdminAccess() throws
    IOException, AccessControlException {
  UserGroupInformation ugi = UserGroupInformation.getCurrentUser();
  UserGroupInformation zkfcUgi = UserGroupInformation.getLoginUser();
  if (adminAcl.isUserAllowed(ugi) ||
    ugi.getShortUserName().equals( zkfcUgi.getShortUserName() )) {
      LOG.info("Allowed RPC access from " + ugi
        + " at " + Server.getRemoteAddress());
      return;
    }
  String msg = "Disallowed RPC access from " + ugi
    + " at " + Server.getRemoteAddress()
    + ". Not listed in " + DFSConfigKeys.DFS_ADMIN;
  LOG.warn(msg);
  throw new AccessControlException(msg);
}
```

| Model | Predictions | |
|---|---|---|
| True ref: | `zkfcUgi.getShortUserName()` | |
| SLM (this work) | **`zkfcUgi.getShortUserName()`** | (11.7%) |
| | `DFSConfigKeys.DFS` | (4.5%) |
| | `zkfcUgi.getUserName()` | (2.6%) |
| Transformer-base +copy | `server.getRemoteAddress()` | |
| | `server.getRemoteUserName()` | |
| | `server.getShortUserName()` | |
| BiLSTM→LSTM +copy | `server.getUserName()` | |
| | `zkfcUgi.getUserName()` | |
| | `ugiUgi.getUserName()` | |
| Seq2tree +copy | `dfsConfigKeys.dfsAdmin` | |
| | `zkfc.getUserName()` | |
| | `zkfcUgi.getRemoteAddress()` | |

Figure 7: Java examples from our test set along with the predictions of our model and the baselines.

```
private C findCounter(T key) {
    int i = key.ordinal();
    if (counters[i] == null) {
        counters[i] = newCounter(key);
    }
    return (C) counters[i] ;
}
```

| Model | Prediction | |
|---|---|---|
| True ref: | `(C) counters[i]` | |
| SLM (this work) | **`(C) counters[i]`** | (71.6%) |
| | `(C) this` | (6.3%) |
| | `counters[i]` | (4.8%) |
| Transformer-base +copy | `(C) this` | |
| | **`(C) counters[i]`** | |
| | `(C) counters` | |
| BiLSTM→LSTM +copy | `(C) this` | |
| | **`(C) counters[i]`** | |
| | `counters[i]` | |
| Seq2tree +copy | **`(C) counters[i]`** | |
| | `(C) counters[i].ordinal()` | |
| | `(C) counters.get(i)` | |

```
private void handleTaskFinishedEvent(TaskFinishedEvent event) {
    TaskInfo taskInfo = info.tasksMap.get( event.getTaskId() );
    taskInfo.counters = event.getCounters();
    taskInfo.finishTime = event.getFinishTime();
    taskInfo.status = TaskStatus.State.SUCCEEDED.toString();
    taskInfo.successfulAttemptId = event.getSuccessfulTaskAttemptId();
}
```

| Model | Prediction | |
|---|---|---|
| True ref: | `event.getTaskId()` | |
| SLM (this work) | `event.getTaskName()` | (8.8%) |
| | `event.getId()` | (8.2%) |
| | `event.getTask()` | (3.4%) |
| Transformer-base +copy | `event.getTaskInfo()` | |
| | **`event.getTaskId()`** | |
| | `event.getId()` | |
| BiLSTM→LSTM +copy | `event.name` | |
| | `event.type` | |
| | `event.getId()` | |
| Seq2tree +copy | `event.getId()` | |
| | `event.getPath()` | |
| | `event.getDescription()` | |

Figure 8: Java examples from our test set along with the predictions of our model and the baselines.

```
static String replaceSubstitution(
    String base, Pattern from, String to, boolean repeat) {
  Matcher match = from.matcher(base);
  if (repeat) {
    return match.replaceAll(to) ;
  } else {
    return match.replaceFirst(to);
  }
}
```

| Model | Prediction | |
|---|---|---|
| True ref: | `match.replaceAll(to)` | |
| SLM (this work) | `match.toString()` | (9.0%) |
| | **`match.replaceAll(to)`** | (8.2%) |
| | `match.replaceAll(to, from)` | (6.5%) |
| Transformer-base +copy | `match.replaceFirst(to)` | |
| | `replace.replaceFirst(to)` | |
| | `matcher.replaceFirst(to)` | |
| BiLSTM→LSTM +copy | `match.getFirst()` | |
| | `match.replaceFirst(to)` | |
| | `match.replaceFirst(to, to)` | |
| Seq2tree +copy | `match.replaceFirst(base)` | |
| | `match.replaceFirst(to)` | |
| | `match.replaceFirst(repeat)` | |

```
public void responseReceived(ResponseReceivedEvent event) {
  RequestResult result = event.getRequestResult();
  Date startDate = result.getStartDate();
  Date stopDate = result.getStopDate();
  long elapsed = stopDate.getTime() - startDate.getTime();
  synchronized (this) {
    this.lastE2Elatency = elapsed;
  }
  if ( LOG.isDebugEnabled() ) {
    int statusCode = result.getStatusCode();
    String etag = result.getEtag();
    HttpURLConnection urlConnection =
        (HttpURLConnection) event.getConnectionObject();
    int contentLength = urlConnection.getContentLength();
    String requestMethod = urlConnection.getRequestMethod();
    long threadId = Thread.currentThread().getId();
    LOG.debug(String.format(
      "SelfThrottlingIntercept:: ResponseReceived:
      ... threadId=%d, Status=%d, Elapsed(ms)=%d,
      ... ETAG=%s, contentLength=%d, requestMethod=%s",
      threadId, statusCode, elapsed, etag, contentLength, requestMethod));
  }
}
```

| Model | Prediction | |
|---|---|---|
| True ref: | `LOG.isDebugEnabled()` | |
| SLM (this work) | `elapsed != null` | (32.1%) |
| | **`LOG.isDebugEnabled()`** | (29.0%) |
| | `!LOG.isDebugEnabled()` | (2.4%) |
| Transformer-base +copy | `stopDate != null` | |
| | `result.hasStatusCode()` | |
| | `result.hasStatusCode() != elapsed` | |
| BiLSTM→LSTM +copy | `result != null` | |
| | `elapsed > 0` | |
| | `result.getStatusCode() == workflowConstants.STATUS` | |
| Seq2tree +copy | `event.getConnectionObject() instanceof HttpUrlConnection` | |
| | `startDate != null` | |
| | **`LOG.isDebugEnabled()`** | |

18

Figure 9: Java examples from our test set along with the predictions of our model and the baselines.

```
private static boolean isNameResolved(InetAddress address) {
  String hostname = address.getHostName() ;
  String ip = address.getHostAddress();
  return !hostname.equals(ip) || NetUtils.isLocalAddress(address);
}
```

| Model | Prediction | |
|---|---|---|
| True ref: | `address.getHostName()` | |
| SLM (this work) | **`address.getHostname()`** | (3.5%) |
| | **`address.getHostName()`** | (2.0%) |
| | `inetAddress.getByName(address.getAddress())` | (0.7%) |
| Transformer-base +copy | `address.getHostAddress()` | |
| | `address.getLastElement().getValue()` | |
| | `address.getAddress()` | |
| BiLSTM→LSTM +copy | `address.getHostAddress()` | |
| | `address.getPort()` | |
| | `address.getAddress()` | |
| Seq2tree +copy | `address.getHostAddress()` | |
| | `address.getPort()` | |
| | `address.getAddress()` | |

```
private synchronized void initJournals(List<URI> dirs) {
  int minimumRedundantJournals = conf.getInt(
      DFSConfigKeys.DFS_NAMENODE_EDITS_DIR_MINIMUM_KEY,
      DFSConfigKeys.DFS_NAMENODE_EDITS_DIR_MINIMUM_DEFAULT);
  journalSet = new JournalSet(minimumRedundantJournals);
  for (URI u : dirs) {
    boolean required =
        FSNamesystem.getRequiredNamespaceEditsDirs(conf).contains(u);
    if ( u.getScheme() .equals(NNStorage.LOCAL_URI_SCHEME)) {
      StorageDirectory sd = storage.getStorageDirectory(u);
      if (sd != null) {
        journalSet.add(
            new FileJournalManager(conf, sd, storage),
            required, sharedEditsDirs.contains(u));
      }
    } else {
      journalSet.add(createJournal(u),
          required, sharedEditsDirs.contains(u));
    }
  }
  if (journalSet.isEmpty()) {
    LOG.error("No edits directories configured!");
  }
}
```

| Model | Prediction | |
|---|---|---|
| True ref: | `u.getScheme()` | |
| SLM (this work) | `u.getName()` | (27.4%) |
| | **`u.getScheme()`** | (13.1%) |
| | `u.getVersion()` | (8.2%) |
| Transformer-base +copy | `journalSet.LOCAL_URI_SCHEME` | |
| | `u.getName()` | |
| | `Boolean.true` | |
| BiLSTM→LSTM +copy | `u.toString()` | |
| | `Boolean.true` | |
| | `u.getURI()` | |
| Seq2tree +copy | **`u.getScheme()`** | |
| | `u.getName()` | |
| | `storage.getLocalUriScheme()` | |

Figure 10: Java examples from our test set along with the predictions of our model and the baselines.

```
static EnumSet<FileAttribute> parse(String s) {
  if (s == null || s.length() == 0) {
    return EnumSet.allOf(FileAttribute.class);
  }
  EnumSet<FileAttribute> set = EnumSet.noneOf(FileAttribute.class);
  FileAttribute[] attributes = values();
  for (char c : s.toCharArray() ) {
    int i = 0;
    for (; i < attributes.length && c != attributes[i].symbol; i++) ;
    if (i < attributes.length) {
      if (!set.contains(attributes[i])) {
        set.add(attributes[i]);
      } else {
        throw new IllegalArgumentException("There are more than one '"
            + attributes[i].symbol + "' in " + s);
      }
    } else {
      throw new IllegalArgumentException("'" + c + "' in "
          + s + " is undefined.");
    }
  }
  return set;
}
```

| Model | Prediction | |
|-------|-----------|---|
| True ref: | `s.toCharArray()` | |
| SLM (this work) | **`s.toCharArray()`** | (22.4%) |
| | `attributes[0].value` | (18.5%) |
| | `attributes[undefined].length` | (4.6%) |
| Transformer-base +copy | `s.split(" "` | |
| | `set.split(" ")` | |
| | `attributes.keySet()` | |
| BiLSTM→LSTM +copy | `attributes.length` | |
| | `attributes[0]` | |
| | `attributes[0].next` | |
| Seq2tree +copy | `set.toArray()` | |
| | **`s.toCharArray()`** | |
| | `set.toCharArray()` | |

```
public static Path[] stat2Paths(FileStatus[] stats) {
  if (stats == null)
    return null;
  Path[] ret = new Path[stats.length];
  for (int i = 0; i < stats.length; ++i) {
    ret[i] = stats[i].getPath() ;
  }
  return ret;
}
```

| Model | Prediction | |
|-------|-----------|---|
| True ref: | `stats[i].getPath()` | |
| SLM (this work) | **`stats[i].getPath()`** | (25.2%) |
| | `new Path(stats[i])` | (3.3%) |
| | `new Path(stats[i], charset)` | (2.5%) |
| Transformer-base +copy | `stats[i]` | |
| | **`stats[i].getPath()`** | |
| | `new Path(stats[i])` | |
| BiLSTM→LSTM +copy | `stats[i]` | |
| | `new Path(stats[i])` | |
| | `stats[i].toString()` | |
| Seq2tree +copy | `stats[i]` | |
| | `new Path(stats[i])` | |
| | `stat(stats[i])` | |

20

Figure 11: Java examples from our test set along with the predictions of our model and the baselines.

```java
void ensureCurrentDirExists() throws IOException {
  for (
      Iterator<StorageDirectory> it = storage.dirIterator();
      it.hasNext(); ) {
    StorageDirectory sd = it.next();
    File curDir = sd.getCurrentDir();
    if ( !curDir.exists()  && !curDir.mkdirs()) {
      throw new IOException("Could not create directory " + curDir);
    }
  }
}
```

| Model | Prediction | |
|---|---|---|
| True ref: | `!curDir.exists()` | |
| SLM (this work) | **`!curDir.exists()`** | (29.0%) |
| | `curDir != null` | (25.8%) |
| | `curDir.exists()` | (24.4%) |
| Transformer-base +copy | `curDir != null` | |
| | **`!curDir.exists()`** | |
| | `curDir.exists()` | |
| BiLSTM→LSTM +copy | `curDir != null` | |
| | `curDir.exists()` | |
| | `sd != null` | |
| Seq2tree +copy | `curDir != null` | |
| | `curDir.exists()` | |
| | **`!curDir.exists()`** | |

```java
public static byte[] getXAttr(final Map<?, ?> json, final String name)
    throws IOException {
  if (json == null) {
    return null;
  }
  Map<String, byte[]> xAttrs = toXAttrs(json);
  if (xAttrs != null) {
    return  xAttrs.get(name) ;
  }
  return null;
}
```

| Model | Prediction | |
|---|---|---|
| True ref: | `xAttrs.get(name)` | |
| SLM (this work) | **`xAttrs.get(name)`** | (28.2%) |
| | `xAttrs.get(xAttrs)` | (5.8%) |
| | `xAttrs.toByteArray()` | (4.4%) |
| Transformer-base +copy | **`xAttrs.get(name)`** | |
| | `xAttrs.toByteArray()` | |
| | `new byte[0]` | |
| BiLSTM→LSTM +copy | `xAttrs.getBytes()` | |
| | `new byte[0]` | |
| | `xAttrs.toByteArray()` | |
| Seq2tree +copy | **`xAttrs.get(name)`** | |
| | `xAttrs.get()` | |
| | `xAttrs.get(0)` | |

Figure 12: Java examples from our test set along with the predictions of our model and the baselines.

```java
private void setFlag(long flag) {
  long prev;
  do {
    prev = unsafe.getLongVolatile(null, this.slotAddress);
    if ( (prev & flag)  != 0) {
      return;
    }
  } while (!unsafe.compareAndSwapLong(
            null, this.slotAddress, prev, prev | flag));
}
```

| Model | Prediction | |
|---|---|---|
| True ref: | `(prev & flag)` | |
| SLM (this work) | **`(prev & flag)`** | (8.9%) |
| | `(prev & flagSlot)` | (5.4%) |
| | `unsafe.get(prev)` | (5.0%) |
| Transformer-base +copy | **`(prev & flag)`** | |
| | `(prev \| flag)` | |
| | `unsafe.compareTo(prev)` | |
| BiLSTM→LSTM +copy | `prev` | |
| | `prev + 1` | |
| | `prev - 1` | |
| Seq2tree +copy | `unsafe prev flag` | (*Syntax error*) |
| | `(volatile prev unsafe.get())` | (*Syntax error*) |
| | `(volatile prev unsafe.getLongVolatile(null, prev))` | (*Syntax error*) |

```java
public synchronized void setInput(byte[] b, int off, int len) {
  if (b == null) {
    throw new NullPointerException();
  }
  if (off < 0 || len < 0  || off > b.length - len) {
    throw new ArrayIndexOutOfBoundsException();
  }
  finished = false;
  if (len > uncompressedDirectBuf.remaining()) {
    this.userBuf = b;
    this.userBufOff = off;
    this.userBufLen = len;
  } else {
    ((ByteBuffer) uncompressedDirectBuf).put(b, off, len);
    uncompressedDirectBufLen = uncompressedDirectBuf.position();
  }
  bytesRead += len;
}
```

| Model | Predictions | |
|---|---|---|
| True ref: | `len < 0` | |
| SLM (this work) | **`len < 0`** | (41.3%) |
| | `off > b.length` | (23.4%) |
| | `len > b.length` | (14.1%) |
| Transformer-base +copy | `off < 0` | |
| | **`len < 0`** | |
| | `b == null` | |
| BiLSTM→LSTM +copy | `off < 0` | |
| | **`len < 0`** | |
| | `b == null` | |
| Seq2tree +copy | `off < 0` | |
| | **`len < 0`** | |
| | `0 < off` | |

Figure 13: Java examples from our test set along with the predictions of our model and the baselines.

```
private int readData(byte[] buf, int off, int len) throws IOException {
  int bytesRead = 0;
  while (bytesRead < len) {
    int n = IOUtils.wrappedReadForCompressedData(
        in, buf, off + bytesRead , len - bytesRead);
    if (n < 0) {
      return bytesRead;
    }
    bytesRead += n;
  }
  return len;
}
```

| Model | Prediction | |
|---|---|---|
| True ref: | `off + bytesRead` | |
| SLM (this work) | `bytesRead - bytesRead` | (35.0%) |
| | **`off + bytesRead`** | (14.1%) |
| | `off - bytesRead` | (9.4%) |
| Transformer-base +copy | `off - bytesRead` | |
| | `off + len` | |
| | `len - bytesRead` | |
| BiLSTM→LSTM +copy | `-bytesRead` | |
| | `bytesRead++` | |
| | `bytesRead - bytesRead` | |
| Seq2tree +copy | `compressed bytesRead` | (*Syntax error*) |
| | **`off + bytesRead`** | |
| | `len - bytesRead` | |

```
private Path getPath(int curId, int limitPerDir, Type type) {
  if (curId <= 0) {
    return basePath;
  }
  String name = "";
  switch(type) {
    case FILE:
      name = FILE_PREFIX + new Integer(curId % limitPerDir).toString();
      break;
    case DIRECTORY:
      name = DIR_PREFIX + new Integer(curId % limitPerDir).toString();
      break;
  }
  Path base = getPath((curId / limitPerDir), limitPerDir, Type.DIRECTORY);
  return  new Path(base, name) ;
}
```

| Model | Prediction | |
|---|---|---|
| True ref: | `new Path(base, name)` | |
| SLM (this work) | **`new Path(base, name)`** | (6.0%) |
| | `new Path(base, name, limitPerDir)` | (2.9%) |
| | `new Path(base, name, type)` | (2.8%) |
| Transformer-base +copy | `new Path(base)` | |
| | `new Path(name)` | |
| | `getPath(base)` | |
| BiLSTM→LSTM +copy | `new Path(base)` | |
| | `new File(base)` | |
| | `new Path(base.getPath())` | |
| Seq2tree +copy | `new Path(base)` | |
| | `new File(base, name)` | |
| | **`new Path(base, name)`** | |

Figure 14: Java examples from our test set along with the predictions of our model and the baselines.

```
private static IEnumerable<Token> OfSequence(
    this IEnumerable<Token> tokens, Token nameToken, TypeDescriptor info)
{
  var nameIndex = tokens.IndexOf(t => t.Equals(nameToken));
  if ( nameIndex >= 0 )
  {
    return info.NextValue.MapValueOrDefault(
        _ => info.MaxItems.MapValueOrDefault(
          n => tokens.Skip(nameIndex + 1).Take(n),
              tokens.Skip(nameIndex + 1).TakeWhile(v => v.IsValue())),
        tokens.Skip(nameIndex + 1).TakeWhile(v => v.IsValue()));
  }
  return new Token[] { };
}
```

| Model | Prediction |
|---|---|
| True ref: | `nameIndex >= 0` |
| SLM (this work) | **`nameIndex >= 0`** (22.6%)
`nameIndex == -1` (19.1%)
`nameIndex > -1` (13.9%) |
| BiLSTM→LSTM +copy | `!nameIndex`
`nameIndex == -1`
`nameIndex < 0` |
| GNN→$\mathcal{NAG}$ (Brockschmidt et al., 2019a) | `nameIndex == 0`
`nameIndex > 0`
`nameIndex < 0` |

```
public static IEnumerable<T[]> Group<T>(
    this IEnumerable<T> source, int groupSize)
{
  if (groupSize < 1)
  {
    throw new ArgumentOutOfRangeException(nameof(groupSize));
  }
  T[] group = new T[groupSize];
  int groupIndex = 0;
  foreach (var item in source)
  {
    group[groupIndex++] = item;
    if ( groupIndex == groupSize )
    {
      yield return group;
      group = new T[groupSize];
      groupIndex = 0;
    }
  }
}
```

| Model | Prediction |
|---|---|
| True ref: | `groupIndex == groupSize` |
| SLM (this work) | `groupIndex < 0` (21.4%)
`groupIndex == -1` (10.3%)
`groupIndex < groupIndex` (5.3%) |
| BiLSTM→LSTM +copy | `group.IsNullOrEmpty()`
`groupGroup[groupIndex++]`
`group.EndsWith(group)` |
| GNN→$\mathcal{NAG}$ (Brockschmidt et al., 2019a) | `groupIndex == 0`
`groupIndex == 1`
**`groupIndex == groupSize`** |

Figure 15: C# examples from our test set of the RestrictC2C task along with the predictions of our model and the baselines.

```
internal static void AddLine(StringBuilder builder,
    string value, int maximumLength)
{
  if (builder == null)
  {
    throw new ArgumentNullException(nameof(builder));
  }
  if (value == null)
  {
    throw new ArgumentNullException(nameof(value));
  }
  if (maximumLength < 1)
  {
    throw new ArgumentOutOfRangeException(nameof(value));
  }

  value =   value.Trim() ;

  builder.AppendWhen(builder.Length > 0, Environment.NewLine);
  do
  {
    var wordBuffer = 0;
    var words = value.Split(' ');
    for (var i = 0; i < words.Length; i++)
    {
      if (words[i].Length < (maximumLength - wordBuffer))
      {
        builder.Append(words[i]);
        wordBuffer += words[i].Length;
        if ((maximumLength - wordBuffer) > 1 && i != words.Length - 1)
        {
          builder.Append(" ");
          wordBuffer++;
        }
      }
      else if (words[i].Length >= maximumLength && wordBuffer == 0)
      {
        builder.Append(words[i].Substring(0, maximumLength));
        wordBuffer = maximumLength;
        break;
      }
      else break;
    }
    value = value.Substring(Math.Min(wordBuffer, value.Length));
    builder.AppendWhen(value.Length > 0, Environment.NewLine);
  }
  while (value.Length > maximumLength);
  builder.Append(value);
}
```

| Model | Prediction | |
|---|---|---|
| True ref: | `value.Trim()` | |
| SLM (this work) | **`value.Trim()`** | (16.0%) |
| | `value.Substring(0, maximumLength)` | (10.9%) |
| | `value.Replace(maximumLength, maximumLength` | (10.7%) |
| BiLSTM→LSTM +copy | `maximumLength - 1` | |
| | **`value.Trim()`** | |
| | `valueLength++` | |
| GNN→$\mathcal{NAG}$ | `value + <UNK>` | |
| | `value + maximumLength` | |
| | `value.Substring(0, maximumLength)` | |

Figure 16: C# examples from our test set of the RestrictC2C task along with the predictions of our model and the baselines.

```
public static string[] TrimStringArray(this IEnumerable<string> array)
{
    return array.Select(item => item.Trim() ).ToArray();
}
```

| Model | Prediction | |
|---|---|---|
| True ref: | `item.Trim()` | |
| SLM (this work) | **`item.Trim()`** | (20.1%) |
| | `item.ToUpperInvariant()` | (3.5%) |
| | `item.ToUpper()` | (1.6%) |
| BiLSTM→LSTM +copy | **`item.Trim()`** | |
| | `item.ToTrim()` | |
| | `item.]` | (*Syntax error*) |
| GNN→NAG (Brockschmidt et al., 2019a) | `item + <UNK>` | |
| | `item + item` | |
| | `item + 1` | |

```
public static string Camelize(this string input)
{
    var word = Pascalize(input);
    return word.Substring(0, 1) .ToLower() + word.Substring(1) ;
}
```

| Model | Prediction | |
|---|---|---|
| True ref: | `word.Substring(0, 1)` | `word.Substring(1)` |
| SLM (this work) | **`word.Substring(0, 1)`** | **`word.Substring(1)`** |
| | `word.Trim()` | `wordData.Substring(1)` |
| | `word.Substring(1)` | `word.Substring(0, 1)` |
| BiLSTM→LSTM +copy | `input.Replace("&", " )` | `input.Replace("&", " <UNK> )` |
| | `input.Replace(1, '')` | `input + "." + input` |
| | `input.Replace("&", "")` | `input.Substring(0, 1)` |
| GNN→NAG | `word.CombineWith(<UNK>)` | `word.CombineWith(<UNK>)` |
| | `word.Trim()` | `word + <UNK>` |
| | `word.CombineWith(input)` | `word.Replace(<UNK>, <UNK>)` |

Figure 17: C# examples from our test set of the RestrictC2C task along with the predictions of our model and the baselines.

```csharp
public string Truncate(string value, int length, string truncationString,
    TruncateFrom truncateFrom = TruncateFrom.Right)
{
  if (value == null)
    return null;

  if (value.Length == 0)
    return value;

  if (truncationString == null)
    truncationString = string.Empty;

  if (truncationString.Length > length)
    return truncateFrom == TruncateFrom.Right ?
      value.Substring(0, length) : value.Substring(value.Length – length);

  var alphaNumericalCharactersProcessed = 0;

  if (value.ToCharArray().Count(char.IsLetterOrDigit) <= length)
    return value;

  if (truncateFrom == TruncateFrom.Left)
  {
    for (var i = value.Length – 1; i > 0; i--)
    {
      if (char.IsLetterOrDigit(value[i]))
        alphaNumericalCharactersProcessed++;

      if (alphaNumericalCharactersProcessed + truncationString.Length
          == length)
        return truncationString + value.Substring(i);
    }
  }

  for (var i = 0; i < value.Length – truncationString.Length; i++)
  {
    if (char.IsLetterOrDigit(value[i]))
      alphaNumericalCharactersProcessed++ ;

    if (alphaNumericalCharactersProcessed + truncationString.Length
        == length)
      return value.Substring(0, i + 1) + truncationString;
  }

  return value;
}
```

| Model | Prediction | |
|---|---|---|
| True ref: | alphaNumericalCharactersProcessed++ | |
| SLM (this work) | **alphaNumericalCharactersProcessed++** | (48.1%) |
| | iCount++ | (5.8%) |
| | iIndex++ | (1.6%) |
| BiLSTM→LSTM +copy | i++ | |
| | truncation++ | |
| | alpha-- | |
| $GNN{\rightarrow}\mathcal{NAG}$ | **alphaNumericalCharactersProcessed++** | |
| | alphaNumericalCharactersProcessed-- | |
| | --alphaNumericalCharactersProcessed | |

Figure 18: C# examples from our test set of the RestrictC2C task along with the predictions of our model and the baselines.

```
public static int BinarySearch<TItem, TSearch>(
    this IList<TItem> list, TSearch value,
    Func<TSearch, TItem, int> comparer)
{
  if (list == null)
  {
    throw new ArgumentNullException("list");
  }
  if (comparer == null)
  {
    throw new ArgumentNullException("comparer");
  }

  var lower = 0;
  var upper = list.Count - 1;

  while (lower <= upper)
  {
    var middle = lower + (upper - lower) / 2;
    var comparisonResult = comparer(value, list[middle]);
    if ( comparisonResult < 0 )
    {
      upper = middle - 1;
    }
    else if ( comparisonResult > 0 )
    {
      lower = middle + 1;
    }
    else
    {
      return middle;
    }
  }

  return lower;
}
```

| Model | Prediction | |
|---|---|---|
| True ref: | `comparisonResult < 0` | `comparisonResult > 0` |
| SLM (this work) | **`comparisonResult < 0`** 
 `comparisonResult > 0` 
 `middle == comparisonResult` | **`comparisonResult > 0`** 
 `comparisonResult < 0` 
 `comparisonResult == 0` |
| BiLSTM→LSTM +copy | `lowerResult == middle` 
 `lowerResult == 0` 
 `lower != middle` | `lower < 0` 
 `lower + "."` 
 `lower != middle` |
| GNN→NAG | `comparisonResult == 0` 
 `comparisonResult > 0` 
 **`comparisonResult < 0`** | `comparisonResult == 0` 
 **`comparisonResult > 0`** 
 `comparisonResult == middle` |

Figure 19: C# examples from our test set of the RestrictC2C task along with the predictions of our model and the baselines.

```
public override string ToString()
{
  // use reflection to display all the properties that
  // ... have non default values
  StringBuilder result = new StringBuilder();
  var props = this.GetType().GetTypeInfo().DeclaredProperties;
  result.AppendLine("{");
  foreach (var prop in props)
  {
    if (prop.Name != "Content" && prop.Name != "Subtitle"
        && prop.Name != "Title" && prop.Name != "UniqueId")
    {
        object value = prop.GetValue(this);
        bool valueIsNull = value == null;
        object defaultValue = Common.GetDefault(prop.PropertyType);
        bool defaultValueIsNull = defaultValue == null;
        if ((valueIsNull != defaultValueIsNull)
            // one is null when the other isn't
          || ( !valueIsNull
              && (value.ToString() != defaultValue.ToString()))))
            // both aren't null, so compare as strings
        {
          result.AppendLine(prop.Name + " : " + prop.GetValue(this));
        }
    }
  }
  result.AppendLine("}");
  return result.ToString();
}
```

| Model | Prediction | |
|---|---|---|
| True ref: | `!valueIsNull` | |
| SLM (this work) | **`!valueIsNull`** | (52.4%) |
| | `!defaultValueIsNull` | (9.0%) |
| | `!valueIsNull.IsNullOrEmpty()` | (3.2%) |
| BiLSTM→LSTM +copy | `!defaultValueIsNull` | |
| | `(defaultValueIsNull || value)` | |
| | `(defaultValueIsNull || defaultValue)` | |
| GNN→NAG (Brockschmidt et al., 2019a) | **`!valueIsNull`** | |
| | `!defaultValueIsNull` | |
| | `!!valueIsNull` | |

Figure 20: C# examples from our test set of the RestrictC2C task along with the predictions of our model and the baselines.

```csharp
public TradierOrderResponse PlaceOrder(string accountId,
    TradierOrderClass classification,
    TradierOrderDirection direction,
    string symbol,
    decimal quantity,
    decimal price = 0,
    decimal stop = 0,
    string optionSymbol = "",
    TradierOrderType type = TradierOrderType.Market,
    TradierOrderDuration duration = TradierOrderDuration.GTC)
{
    //Compose the request:
    var request = new RestRequest("accounts/{accountId}/orders");
    request.AddUrlSegment("accountId", accountId.ToString());

    //Add data:
    request.AddParameter("class", GetEnumDescription(classification));
    request.AddParameter("symbol", symbol);
    request.AddParameter("duration", GetEnumDescription(duration));
    request.AddParameter("type", GetEnumDescription(type));
    request.AddParameter("quantity", quantity);
    request.AddParameter("side", GetEnumDescription(direction));

    //Add optionals:
    if (price > 0) request.AddParameter("price", Math.Round(price, 2));
    if (stop > 0) request.AddParameter("stop", Math.Round(stop, 2));
    if ( optionSymbol != "" )
        request.AddParameter("option_symbol", optionSymbol);

    //Set Method:
    request.Method = Method.POST;

    return Execute<TradierOrderResponse>(request,
        TradierApiRequestType.Orders);
}
```

| Model | Prediction | |
|---|---|---|
| True ref: | `optionSymbol != ""` | |
| SLM (this work) | **`optionSymbol != ""`** | (5.5%) |
| | `optionSymbol == ""` | (4.4%) |
| | `optionSymbol.IsNullOrEmpty()` | (1.1%) |
| BiLSTM→LSTM +copy | `!stopSymbol` | |
| | `stopSymbol != optionSymbol` | |
| | `(stopSymbol " && optionSymbol)` | (*Syntax error*) |
| GNN→NAG (Brockschmidt et al., 2019a) | `optionSymbol == <UNK>` | |
| | `optionSymbol == symbol` | |
| | `optionSymbol != symbol` | |

Figure 21: C# examples from our test set of the RestrictC2C task along with the predictions of our model and the baselines.

```
[Test, TestCaseSource("GetLeanDataLineTestParameters")]
public void GetSourceMatchesGenerateZipFilePath(
    LeanDataLineTestParameters parameters)
{
    var source = parameters.Data.GetSource(
        parameters.Config, parameters.Data.Time.Date, false);
    var normalizedSourcePath = new FileInfo(source.Source).FullName;
    var zipFilePath = LeanData.GenerateZipFilePath(
        Globals.DataFolder, parameters.Data.Symbol,
        parameters.Data.Time.Date,
        parameters.Resolution, parameters.TickType);
    var normalizeZipFilePath = new FileInfo(zipFilePath).FullName;
    var indexOfHash = normalizedSourcePath.LastIndexOf(
        "#", StringComparison.Ordinal);
    if (indexOfHash > 0)
    {
        normalizedSourcePath =
            normalizedSourcePath.Substring(0, indexOfHash) ;
    }
    Assert.AreEqual(normalizeZipFilePath, normalizedSourcePath);
}
```

| Model | Prediction | |
|---|---|---|
| True ref: | `normalizedSourcePath.Substring(0, indexOfHash)` | |
| SLM (this work) | **`normalizedSourcePath.Substring(0, indexOfHash)`** | (28.3%) |
| | `normalizedSourcePath.Substring(1)` | (8.8%) |
| | `normalizedSourcePath.Remove(indexOfHash)` | (8.2%) |
| BiLSTM→LSTM +copy | `indexOfHash + "<UNK>"` | |
| | `indexOfHash > normalizedOfHash` | |
| | `indexOfHash > 0` | |
| GNN→$\mathcal{NAG}$ | `normalizedSourcePath + normalizeZipFilePath` | |
| | `normalizedSourcePath + normalizedSourcePath` | |
| | `normalizedSourcePath + normalizeZipFilePath + <UNK>` | |

Figure 22: C# examples from our test set of the RestrictC2C task along with the predictions of our model and the baselines.

