# OpenReview forum: "Structural Language Models for Any-Code Generation"
_ICLR.cc/2020/Conference — Reject_

### Official Review · AnonReviewer2 · 2019-10-23
**Official Blind Review #2**

**Rating:** 6

**Review:**

The paper proposes a model to address the Any-Code Generation (AnyGen) task, which basically to fill missing code from a given program. The model makes use of partial Abstract Syntax Tree (AST) as input. The model learns representation for partial AST paths and use the learnt representation to generate AST node at masked steps. The conducted experiments show that using AST paths from root and leaves are good for AST node generation, but whether those inputs are robust and sufficient should be further explored.

There are some restrictions to the method, for example,  the input is only a single function, and the missing expression is not that complex. Nevertheless this work presents a novel method towards code generation. The paper also introduces a new metric to evaluate the prediction accuracy for generated expressions. Writing is clear. Evaluation is fairly comprehensive.

Questions:
1. Did the author test the method without the camel notation assumption, i.e. the data contains non-camel notation or mixed notations?
2. In the Restrict Code Generation test, it seems that the author filters out non-primitive types and user-defined functions. Therefore, does the experiment on Java-small dataset fully show the proposed model’s strength?
3. Can the author explain why the SLM model fail on Figure 6? Is it because of dividing token into sub tokens?
4. How big is the token vocabulary? How does the vocab size affect the performance?


**Experience Assessment:**

I have read many papers in this area.

**Review Assessment: Checking Correctness Of Derivations And Theory:**

I assessed the sensibility of the derivations and theory.

**Review Assessment: Checking Correctness Of Experiments:**

I assessed the sensibility of the experiments.

**Review Assessment: Thoroughness In Paper Reading:**

I read the paper thoroughly.

---

> ### Author Response · Authors · 2019-11-10
> **Authors Response to AnonReviewer2**
>
> Thank you for your kind and detailed review.
> You raise many important points, which we think are all addressable within the discussion phase. Please see our detailed response below.
> We are glad that you are convinced that our work "presents a novel method towards code generation". If there are additional questions or experiments that we can perform to improve the paper and improve your rating - please let us know.
>
> | the input is only a single function
> This is correct, but this is just for the sake of practicality and controlled experimentation; this is not a limitation of our approach.
> Our model can be easily trained on additional information from other functions, other classes, or other files in the project, depending on the practical requirements.
>
> | the missing expression is not that complex
> Since we took any expression, the number of possibilities is exponential and the task is extremely difficult. As demonstrated in the Appendix, these expressions are very hard for a human to predict. There is no other currently existing approach that performs better, thus our work is an important step.
>
> | 1. Did the author test the method without the camel notation assumption, i.e. the data contains non-camel notation or mixed notations?
> Yes. In our preliminary experiments, we experimented with not splitting tokens to subtokens at all, and the results were degraded by about 50%. The reason is that identifier names were much more sparse; the model could not copy subtokens; and many of the tokens were treated as UNKs (i.e., the complete name "FileStatus" is UNK, although both "file" and "status" are in the vocabulary).
> In our final proposed model, we split based on "camelCase", "PascalCase", and "snake_case" during preprocessing.
>
> | 2. In the Restrict Code Generation test, it seems that the author filters out non-primitive types and user-defined functions.
> | Therefore, does the experiment on Java-small dataset fully show the proposed model’s strength?
> In general, we address the "Any Code to Code Generation" task (without limitations on types and functions). This is the task that we evaluated on Java-small, and this experiment *does* fully show the proposed model's strength.
> The purpose of evaluating our model on the "RestrictGen" task was only to provide a fair comparison to Brockschmidt et al. (2019), who released only a RestrictGen model, and only in C#.
> So, we filtered out non-primitives and user-defined functions only in C#, because otherwise, the comparison would not have been fair towards the baseline. The "AnyGen" (will be changed to "AnyC2C" as per suggestions) evaluation in Java is the main task that shows the model's strength.
>
> | 3. Can the author explain why the SLM model fail on Figure 6? Is it because of dividing token into sub tokens?
> Without dividing tokens into subtokens, predicting the ground truth will not even be possible, because the name "getTaskId" is not common and does not appear in context (thus not copyable).
> Thanks to splitting to subtokens, our model predicted the ground truth ("event.getTaskId()") as the 5th candidate (not shown in the figure) with a probability of 3.3%.
> As the 1st candidate, our model predicted "event.getTaskName()".
> Generating the correct identifier ["get","task","id"] is just very difficult since neither "getTaskId" nor "TaskId" appears in the context and there is no apparent hint for them.
> In this case, the problem is just extremely difficult, and only feasible thanks to splitting to subtokens.
>
> | 4. How big is the token vocabulary? How does the vocab size affect the performance?
> We did not tune the vocabulary size much, but we did find in our Java experiments that a vocabulary of size 1000 produces better results than vocabulary of size 20,000 and 30,000. We think that by using oversized vocabularies the model learns to overfit the specific target names in the training data, and to rely less on the copy mechanism. By limiting the model to a smaller vocabulary, the model learns to rely more on copying from the test context rather than predicting from the training vocabulary.

---

### Official Review · AnonReviewer3 · 2019-10-24
**Official Blind Review #549**

**Rating:** 6

**Review:**

This paper proposes a generative task for programming code where an expression from the program is generated given the rest of the program (minus the expression). This is in line with language modeling for natural language. The proposed method generates the AST corresponding to the program by generating one node at the time for the missing/to-be-generated expression by approximating the probabilities of the generated notes. Again, this is similar to the prediction of words in language models.

The method takes into account the AST corresponding to the program. However, when representing the program, the structure of the AST is not preserved, instead, the AST is represented by generating several sequential paths by traversing paths between connected nodes in the tree.

It would be nice if the paper provided some intuition why generating such connecting paths in the tree are relevant for representing the code, specially for nodes that do not have a direct relationship between them (e.g., the nodes are distant enough in the code that the corresponding probabilities of their nodes do not seem/appear related).

The paper presents results for 2 datasets (comparing with various related work methods). The results for the Java dataset improve state of the art by 1-2%, while the results for the restricted C# dataset show a much more significant improvement (in the order of 10-15% improvement, depending on the metric).

I would have liked to see a qualitative analysis of the results. In particular, I would have liked to understand how the predictions differ between acc and tree metrics. In other words, when the prediction looking at the tree structure is correct and the overall prediction is not, what goes wrong?

It was not clear to me why or if all the paths between 2 nodes are necessary when encoding the partial AST and predicting the missing nodes. I was not convinced that the ablation studies were relevant. I would have liked to see ablation studies that considered a subset of the paths in the graph.

The elimination of the methods with more than 20 lines of code seems ad-hoc to me and biases the evaluation with relatively short methods (how many methods were eliminated this way?).

One thing that I struggle with is understanding how useful the proposed task is and how it can be generalized/used in practice for some relevant higher level task in AI4code.


**Experience Assessment:**

I have published one or two papers in this area.

**Review Assessment: Checking Correctness Of Derivations And Theory:**

I assessed the sensibility of the derivations and theory.

**Review Assessment: Checking Correctness Of Experiments:**

I carefully checked the experiments.

**Review Assessment: Thoroughness In Paper Reading:**

I read the paper thoroughly.

---

> ### Author Response · Authors · 2019-11-10
> **Authors Response to AnonReviewer3 (1/2)**
>
> Thank you for your kind and detailed review. You raise many important points, which we think are all addressable within the discussion phase. Please see our detailed response below.
> If there are additional questions or experiments that we can perform to improve the paper and improve your rating - please let us know.
>
> | how useful the proposed task is and how it can be generalized/used in practice?
> Our approach models code, just like LMs model natural language. If we look at how LM research has contributed to NLP in the past years (NMT, BERT, etc.), it is very natural to draw the analogy to our work and project how researching generic modeling of code can enable future programming language processing applications.
>
> Having said that, there are also immediate applications of our work, for example:
> 1. As a code completion assistant in the developer's IDE, which can generate complete expressions and statements, rather than a single token at a time as existing tools.
> 2. As an "automatic code review", by "hiding" existing pieces of code and asking the model to predict them, thus detecting unlikely pieces of code (that may contain bugs or just be confusing).
>
> Overall, the task addressed in this paper is the closest task to the realistic settings of a programmer who writes code and can be aided by a neural model, and is certainly more realistic than the existing restricted benchmark of Brockschdmidt et al. (2019).
>
> | The elimination of the methods with more than 20 lines of code seems ad-hoc to me and
> | biases the evaluation with relatively short methods (how many methods were eliminated this way?).
> About 10% of the examples were eliminated this way (the average method has about 7 lines). In preliminary experiments, we saw that this filtering actually helps the seq2seq baselines no less than it helped our model. Even under 20 lines of code - some examples were more than 1000 subtokens long, thus are very difficult for both LSTMs and Transformers to learn. Without the 20-lines limitation - some examples were over 5000 subtokens long.
> Methods longer than 20 lines were frequently tests, configurations, initializations, or auto-generated code.
>
> | It would be nice if the paper provided some intuition why generating such connecting paths in the tree are relevant for representing
> | the code especially for nodes that do not have a direct relationship between them (e.g., the nodes are distant enough in the code ... )
> Intuitively, an AST path is a relationship between nodes where the kind of the relationship is  *known and unambiguous* (in contrast to natural language). For example, an AST path is a simple way to connect a variable declaration to its usage (regardless of their textual proximity).
> The set of paths provide the model with information about all other symbols that surround the prediction location, along with the syntactic relation to each symbol. These long-distance syntactic relations provide a good approximation of the available information in context. Additionally, AST paths facilitate learning because they normalize much of the textual-form of code, focusing on syntactic patterns rather than textual patterns (as observed by Alon et al., ICLR'2019).
>
> For example, in Figure1(left) - there exists a path from the variable "i" (in "int i = 0") into the highlighted green box. Intuitively, this path captures the fact that  "i" is the iterator index of the current "for" loop (hinting that it should be used somehow, as the "i" index is usually used inside "for" loops). There are additional paths from the parameter declaration "FileStatus[] stats" that capture the fact that "stats" is an array of type FileStatus. Additionally, there is no path from another occurrence of "stats" that show that "stats" have been accessed, except for its ".length" field.
> All these paths signal the model most of the relevant information in this case: "we have not used the array 'stats' yet, and we probably need to also use the index 'i'", leading to generate "stats[i]" first.
>
> It is very difficult to tell in advance which paths will be useful. See, for example, Figure 9 (top, page 18): the prediction of "s" comes from the declaration of "String s" which is 6 lines above.
> However, if a path is completely useless, the model will (hopefully) learn to give it less attention.

---

> > ### Author Response · Authors · 2019-11-10
> > **Authors Response to AnonReviewer3 (2/2)**
> >
> > | It was not clear to me why or if all the paths between 2 nodes are necessary when encoding the partial AST and predicting the missing nodes.
> > | I would have liked to see ablation studies that considered a subset of the paths in the graph.
> > Choosing the right subset in advance is an open question, which could be a really interesting direction for future work. In our preliminary experiments, we did not find a good way of doing that. Any manual limitation of paths reduced the accuracy.
> > This is consistent with Alon et al. (2018) who introduced the AST paths and studied the performance (on different tasks than ours) when changing the lengths and widths of the paths. As observed in the experiments of Alon et al. (2018), having more paths, and longer paths (more information) improved accuracy.
> > Additionally, processing an arbitrarily large number of paths is not difficult computationally, since many paths can be batched and processed in parallel on GPUs.
> > If you have a specific ablation in mind we would love to run it.
> >
> > | I would have liked to see a qualitative analysis of the results ...
> > | When the prediction looking at the tree structure is correct and the overall prediction is not, what goes wrong?
> > Usually, we get only a partially correct assignment of subtokens. See for example Figure 6 (bottom, page 15). Our model predicts "event.getTaskName()" which is "tree-equal" to the ground truth: "event.getTaskId()".  Generating the correct identifier ["get","task","id"] is very difficult since "TaskId" doesn't appear as-is in the context and there is no apparent hint.
> >
> > Note that this task is very difficult, and most of the time impossible without having the entire project and its dependencies in context and knowing the developer's intent.
> > We will add a qualitative analysis of this example and others in the next revision (to be uploaded today).
> >
> > | The results for the Java dataset improve state of the art by 1-2%
> > This improvement is significant, as it is compared to very strong state-of-the-art models that we did our best to strengthen and tune across multiple runs, including subtokenization, copying, and adding more learnable parameters than in our models.

---

### Official Review · AnonReviewer4 · 2019-10-30
**Official Blind Review #4**

**Rating:** 1

**Review:**

This paper presents a grammar-based generation approach for "slot-filling" style code generation tasks. Given a context AST with opening non-terminal node, the model completes the opening node by predicting a sequence of child nodes, which forms a sub-AST rooted at the original opening node. The proposed model encodes context ASTs using a path-based approach (Alon et al., 2019a), essentially generalizing the previous model of Alon et al., 2019a from a code-to-sequence setting (e.g., generating natural language comments from code) to a "code-to-code" setting (i.e., code completion given contextual snippets).

Strong Points:

* The paper is very well written. The idea of formalizing code completion as structured language modeling and extending Alon et al., 2019a for the task is natural and well executed, with strong models and significantly improved results on two code completion benchmarks for both Java and C#.

* The authors attempted to establish comparisons with most existing code generation models.

Detailed Review:

*Technical Contribution* I have a very mixed feeling with this paper, while the model registers high empirical performance, the technical contribution is a bit limited, as detailed below:

    - *Path-based Context Encoding* The most important contribution in this submission is the application of path-based AST encoding model of Alon et al., 2019a to encode context (the given contextual and partially generated ASTs) for code generation. While the path-based encoding scheme is indeed a powerful model that intuitively encapsulates and generalizes over most previous approaches (Section 7), applying the model to a different task without significant task-specific adaptation or in-depth analysis might not sound technically novel. Meanwhile, the core idea of modeling/encoding the information flow in both the given context AST and partially generated programs for opening node expansion has already been explored in Brockschmidt et al. (2019a), albeit using a different encoding approach (GNNs) and in a relatively restricted setting of generating arithmetic expressions.

    - *Node-based Tree Generation Model* Apart from the path-based context encoding model, the node-based generation model presented in Section 2 also seems interesting. However, it might take longer time-steps to generate the node sequence instead of the sequence of production rules (composed of multiple child nodes), which could make optimization and inference more difficult. On the other hand, to control arity, the node-based approach need to inject synthetic "EOS" nodes to signal end of generating an argument sequence, while existing production rule-based systems could easily generate arbitrary number of argument nodes using either a transition system (e.g., Yin and Neubig, 2018) or a special neural component to compute end-of-argument-list probability (e.g., Rabinovich et al. (2017)), without using separate production rules of different arity.

    - *Syntactic Copy Mechanism* While the proposed syntactic terminal token copy mechanism (Section 3.3) could be better than the vanilla sequential one, there have already been syntactic copying models capable of copying both terminal tokens and partial ASTs from the context (Yin et al., 2019).

    How to Improve: to better understand the different technical contributions outlined above and their relative impacts, the following ablation studies would be helpful:

        - Importance of Path-based Context Encoding: the Seq→Path ablation in Table 3 alone might not be adequate to demonstrate the importance of path-based encoding of AST contexts for code generation tasks. The authors should compare with the GNN-based context encoding approach in Brockschmidt et al. (2019a) as this is the most relevant work. The original GNN→NAG model cited in Table 2 used a much simpler copying mechanism and a vanilla production-based tree generation model, and therefore not directly comparable with a tuned SLM.

        - Importance of Node-based Tree Generation Model: If possible, the authors might consider swapping their node-based tree generation model with a state-of-the-art production-based approach (e.g., Yin et al., 2019) to demonstrate its effectiveness.

*Claims* The authors claimed in the beginning of the paper that previous program synthesis approaches are either restricted in domains (e.g., DSLs like SQL) or grammars (e.g., restricted grammar of the full language), therefore coining the proposed approach as "any-code generation". However, there are indeed code generation systems (some of them cited in this paper) that synthesize open-domain code snippets in general-porpuse programming languages without restriction on vocabulary or grammar. To give a few examples, Iyer et al. (2018) generate open-domain Java class member functions; Rabinovich et al. (2017) and Zhao et al. (2019) predict full Python classes or partial snippets, while Yin et al. (2019) synthesize open-domain short C# code diffs observed in GitHub commits. In fact, the the proposed "any-code generation" benchmark is limited to sub-expressions defined within a function, whose scope is more restricted than other benchmarks like CONCODE.

    How to improve: the authors might present more evidence to substantiate the their claim on the novelty of the AnyGen benchmark compared with existing open-domain, general purpose code synthesis benchmarks, or consider revising the claim and the title.

References:

* Zhao et al., Neural Networks for Modeling Source Code Edits. 2019
* Yin et al., Learning to Represent Edits. 2019


**Experience Assessment:**

I have published in this field for several years.

**Review Assessment: Checking Correctness Of Derivations And Theory:**

I assessed the sensibility of the derivations and theory.

**Review Assessment: Checking Correctness Of Experiments:**

I carefully checked the experiments.

**Review Assessment: Thoroughness In Paper Reading:**

I read the paper thoroughly.

---

> ### Author Response · Authors · 2019-11-10
> **Authors Response to AnonReviewer4 (1/2)**
>
> Thank you for your kind words and for your detailed review. You raise many important points, which we think are all addressable within the discussion phase. Please see our detailed response below.
>
> | "Any-Code Generation" was already presented in prior work. The paper would be improved if the claim and
> | the title would be revised
> We agree, this term could be incorrectly interpreted too widely.
> To clarify that our "any code generation" is limited to the "code generation given the surrounding code" task - we will remove the word "any" from the title (resulting in: "Structural Language Models for Code Generation") and change the term "AnyGen" to the more specific name "AnyC2C" for "Any Code-to-Code". Our work is certainly not the first to perform "any code generation", but only to perform "any code generation given the surrounding code". We are aware that prior work performed "any code generation" for NL->code (as Rabinovitch et al. (2017), Yin and Neubig (2017), Iyer et al. (2018)) and generating any code edits (Yin et al. 2019). We will make sure that all of these are cited appropriately.
>
> If this was your main concern, and you are convinced that our idea is "well executed" and "generalizes over most previous approaches", that our approach "registers high empirical performance", and has "significantly improved results", will you consider recommend accepting our paper?
>
> | Syntactic copy mechanism has already been presented in Yin et al. (2019)
> We agree. We will refer to Yin et al. (2019) in this context, and focus on describing our syntactic copy mechanism.
>
> | "The node-based generation ... might take longer time-steps to generate the node sequence ... than production
> | rules. The node-based approach needs to inject synthetic "EOS" nodes to signal end of generating an argument
> | sequence, while existing production rule-based systems could easily generate arbitrary number of argument
> |nodes using either a transition system..."
> We agree that existing systems could generate arbitrary number of arguments as well, we will include a reference to Yin & Neubig (2018) and Rabinovich (2017) in this context.
>
> That being said, note that our node-level generation is different and clearly novel. It might take more time steps than rule-level generation, but decomposing the structured prediction into more atomic steps can also be helpful, as the model does not need to commit to a large decision. This decomposition allows the model to commit to a "higher confidence" small decision first, and make the next predictions based on the earlier steps. Injecting the "synthetic" EOS nodes is not a problem and does not add a significant burden; it is just another orthogonal dimension for the EOS nodes that already exist in rule-level models and in seq2seq models. Our average number of targets is 10.8; for the seq2seq baselines the average is 7.8 targets; if we modeled our targets using production rules, the average number of targets would have been 7.9. This question is also related to byte pair encoding (BPE) in NMT: BPE increases the number of tokens but usually improves results.
>
> Another significant advantage of the node-level generation is that it allows to jointly learn the context and the prediction, leading to better generalization: each AST path goes through both context nodes and previously-predicted nodes. This joint modeling would have been much more complicated to perform in rule-level models.
>
> This is basically an empirical question, and as you stated, our model "registers high empirical performance". In all of our experiments, this decomposition did not show any significant optimization or inference difficulties over the baselines.
>
> | "The most important contribution in this submission is the application of path-based AST encoding model
> | of Alon et al., 2019a to encode context ... While the path-based encoding scheme is indeed a powerful
> | model that intuitively encapsulates and generalizes over most previous approaches, applying the
> | model to a different task without significant task-specific adaptation or in-depth
> | analysis might not sound technically novel"
> The most important contribution in our paper is the novel structural language model for *generating* code; encoding context using AST paths was the focus of Alon et al., (2019a). Jointly encoding the context and the prediction as a Structural Language Model is the main contribution of our paper. All previous work presented encoder-decoder models (possibly because the input was different). Our paper is the first to address this problem by jointly modeling the context and the prediction using a structural language model.
>
> Furthermore, generating code is not just a different task, but a completely different structured prediction problem than generating a natural language sequence. Thus, our "code-to-code" task is much more challenging than an "application to a different task" of code2seq (Alon et al., 2019a).

---

> > ### Author Response · Authors · 2019-11-10
> > **Authors Response to AnonReviewer4 (2/2)**
> >
> > | "the core idea of modeling/encoding the information flow in both the given context AST
> > | and partially generated  programs for opening node expansion has already
> > | been explored in Brockschmidt et al. (2019a)"
> > This is correct, but as you stated: Brockschmidt et al (2019) performed it in a "restricted setting" and our model presents "significantly improved results". The main conceptual difference between our work and Brockschmidt et al. (2019) is our observation that this task should be addressed with a *joint structural language model, rather than a separate encoder and decoder*. This observation might not even be applicable in the model of Brockschmidt et al. (2019). This observation is applied in our model using AST paths to jointly model context and prediction, while in their model they use a GNN encoder for the context and a NAG rule-level decoder.
> >
> > Additionally, the model of Brockschmidt et al. (2019) cannot be easily applied to "any code" as our model, because names and arity of invoked functions are hard-coded in their production rules (see for example the paragraph "Node Trees vs. Production Trees" in Section 2, which contains a specific comparison to their model).
> >
> > | Path-based context encoding compared to GNN-based context encoding - "The authors should compare
> > | with the GNN-based context encoding approach in Brockschmidt et al. (2019a) as this is the most relevant
> > | work. The original GNN→NAG model cited in Table 2 used a much simpler copying mechanism and a
> > | vanilla production-based tree generation model
> > We compared our model to GNN->NAG (Brockschmidt et al. (2019a)) using their original implementation, as this is indeed the most relevant work and only existing implementation for this task. Their implementation actually incorporated additional improvements for the copying mechanism that were not used in their paper, such as ideas from Cvitkovic et al. (ICML'2019), so their copying mechanism is not that simple. Tree generation using "Neural Attribute Grammar" ("NAG") is the main contribution of their paper, so we are finding it hard to call it a "vanilla production-based tree generation". They also found their NAG tree generation to perform much better than vanilla production-based tree generation approaches (called "Tree" and "ASN" in their paper) and better than "Syn" which follows Yin & Neubig (2017).
> >
> > In general, we found that the main point in this "code-to-code" task is that instead of optimizing the choices of "the best encoder" and "the best decoder" separately - the most important thing is to use the *same type of encoder and decoder*, hopefully jointly as the same component (by tying their weights). Using AST paths only in either the encoder or the decoder is not enough - AST paths are useful mostly when used to encode the entire tree (context *and* target subtree), as shown in the ablation study.
> >
> > Comparing different context encoding approaches is not quite the focus of this paper, as we focus on *generating code* by jointly learning it with the context.
> > The different context-encoding approaches are shown in "code2seq" (Alon et al. 2019) and "graph2seq" (Fernandes et al, 2019): when fixing the same LSTM decoders without copying, Fernandes et al. (2019) showed that AST paths encode context better than GNNs, unless when the GNN is ensembled with an LSTM token-based encoder (i.e., "BiLSTM+GNN -> LSTM").
> >
> > | the authors might consider swapping their node-based tree generation model with a state-of-the-art
> > | production-based approach (e.g., Yin et al., 2019) to demonstrate its effectiveness
> > Yin et al. (2019) addressed a completely different task of encoding code edits and applying the encoded edit on a new code, which is very different than our settings. As far as we know, Yin et al. (2019) did not release their model code, but only code that scrapes data. We will include a reference to Yin et al. (2019) in this context and discuss their rule-level approach. Note that for the code-to-code task, Brockschmidt et al. (2019) have implemented state-of-the-art production-level tree generation approach, and we compared to their implementation.
> > Keep in mind that swapping our tree generation model (to anything else) will not allow modeling the context tree jointly with the target subtree - which is the main idea of our paper and an important contributing factor.

---

### Author Response · Authors · 2019-11-10
**Submission update 11/10/2019 - Summary of changes**

We would like to thank the reviewers for all their excellent suggestions!
We updated our submission to address the following comments raised by the reviewers:

(1) Following the suggestions of AnonReviewer4, and to emphasize that our "any code generation" is limited to the "code generation given the surrounding code" task, we:
    (1A) Removed the word "Any" from the title, resulting in "Structural Language Models for Code Generation".
    (1B) We changed the term "AnyGen" to the more specific name "AnyC2C" (Any Code-to-Code).
We are aware that previous work also performed general-purpose code generation.  We claim "any generation" only in the "code-context to code" task and *not in* "NL-to-code" nor "Edit-to-code" tasks of prior work.

(2) To highlight the importance of structural language modeling (AnonReviewer4) - we included an additional ablation of "Paths2Paths", in which the encoder and decoder are both path-based, but have untied weights. This ablation shows that the scores of the following models are ordered in the following order:
                                                      SLM > Paths2Paths > seq2seq > Paths2seq > Seq2Path
    (2A) As SLM performs better than Paths2Paths, this ablation shows the importance of joint modeling of the context and the target subtree by parameter tying.
    (2B) Each of Paths2Paths and seq2seq performs better than Paths2seq and Seq2path; this shows the importance of *using the same type of encoder and decoder* for the code-context-to-code task, rather than combining "an optimal encoder" with "an optimal decoder".
    (2C) Paths2Paths performs better than seq2seq, showing the advantage of using paths over textual sequences.

(3) We included a qualitative analysis (new Section 7). (AnonRev3, AnonRev2)

(4) We added citations and references to Yin et al. (ICLR 2019), mentioning their Tree-copy mechanism and "any code generation" in the context of edits (AnonRev4).

(5) We added citations and references to Yin & Neubig (EMNLP 2018), mentioning their ability to produce an arbitrary number of arguments using a transition system (AnonRev4). This work is indeed very relevant. Thank you AnonReviewer4 for mentioning it.

(6) We added references to Rabinovich et al (ACL'2017), mentioning their "horizontal LSTM" to produce an arbitrary number of child nodes (AnonRev4).

(7) We added citations and references to Cvitkovitc et al. (ICML'2019), as the implementation of the GNN->NAG baseline includes ideas from Cvitkovitc et al.

---

### Decision · Program_Chairs · 2019-12-19

**Decision:**

Reject

**Comment:**

This paper proposes a new method for code generation based on structured language models.

After viewing the paper, reviews, and author response my assessment is that I basically agree with Reviewer 4. (Now, after revision) This work seems to be (1) a bit incremental over other works such as Brockschmidt et al. (2019), and (2) a bit of a niche topic for ICLR. At the same time it has (3) good engineering effort resulting in good scores, and (4) relatively detailed conceptual comparison with other work in the area. Also, (5) the title of "Structural Language Models for Code Generation" is clearly over-claiming the contribution of the work -- as cited in the paper there are many language models, unconditional or conditional, that have been used in code generation in the past. In order to be accurate, the title would need to be modified to something that more accurately describes the (somewhat limited) contribution of the work.

In general, I found this paper borderline. ICLR, as you know is quite competitive so while this is a reasonably good contribution, I'm not sure whether it checks the box of either high quality or high general interest to warrant acceptance. Because of this, I'm not recommending it for acceptance at this time, but definitely encourage the authors to continue to polish for submission to a different venue (perhaps a domain conference that would be more focused on the underlying task of code generation?)